# Electrothermally controlled origami fabricated by 4D printing of continuous fiber-reinforced composites

Yaohui Wang [1,4], Haitao Ye[2,3,4], Jian He [1], Qi Ge [2] ✉ & Yi Xiong [1] ✉

Active origami capable of precise deployment control, enabling on-demand modulation of its properties, is highly desirable in multi-scenario and multi-task applications. While 4D printing with shape memory composites holds great promise to realize such active origami, it still faces challenges such as low load-bearing capacity and limited transformable states. Here, we report a fabrication-design-actuation method of precisely controlled electrothermal origami with excellent mechanical performance and spatiotemporal controllability, utilizing 4D printing of continuous fiber-reinforced composites. The incorporation of continuous carbon fibers empowers electrothermal origami with a controllable actuation process via Joule heating, increased actuation force through improved heat conduction, and enhanced mechanical properties as a result of reinforcement. By modeling the multi-physical and highly nonlinear deploying process, we attain precise control over the active origami, allowing it to be reconfigured and locked into any desired configuration by manipulating activation parameters. Furthermore, we showcase the versatility of electrothermal origami by constructing reconfigurable robots, customizable architected materials, and programmable wings, which broadens the practical engineering applications of origami.

Active origami is an emerging subset of origami with the ability to alter its configuration and characteristics, such as geometrical[1,2], mechanical[3,4], or electromagnetic[5] properties, in response to external stimuli. This shape-shifting capability is achieved by utilizing stimulus-responsive materials[6], obviating the need for external mechanical loads and bulky actuation systems, such as motors[7–9] and pneumatic pumps[10–12]. The self-actuation bestowed by its all-in-one nature makes active origami particularly intriguing to applications with extreme conditions, e.g., confined space and limited weight, as evident in microdevices[13,14], foldable robots[15,16], and aerospace systems[17]. However, fabricating active origami remains challenging, particularly with the conventional multi-step approach,

due to its intricate crease patterns and spatial distributions of responsive/passive materials.

4D printing[18] is a compelling single-step method for constructing active origami with integrated stimulus-responsive shape-shifting behavior endowed by 3D printing structures with responsive materials, whose cost is insensitive to the complexity of crease layout and material distributions. Several instances of origami have been successfully crafted via 4D printing employing shape memory polymer (SMP), particularly the thermal responsive kind, owing to its high technological readiness[19–22]. However, SMP-based active origami suffers from low stiffness, especially in high temperatures, which significantly limits its load-bearing ability. Additionally, the

[1]School of System Design and Intelligent Manufacturing, Southern University of Science and Technology, Shenzhen 518055, China. [2]Department of Mechanical and Energy Engineering, Southern University of Science and Technology, Shenzhen 518055, China. [3]Department of Mechanical Engineering, City University of Hong Kong, Kowloon, Hong Kong SAR, China. [4]These authors contributed equally: Yaohui Wang, Haitao Ye. ✉e-mail: geq@sustech.edu.cn; xiongy3@sustech.edu.cn

environmental stimulus, e.g., via water bath[23] or oven heating[24], transforms origami only between two states and cannot selectively activate specific hinges, leaving vast unexplored configurations and untapped potential. Previous efforts to tackle these challenges have used composite responsive materials filled with carbon nanofibers[25]/nanotubes[26,27], magnetic particles[28], and photoabsorbers[29]. Yet, the enhancements in mechanical performance remain marginal. Meanwhile, the inconsistent dispersion of conductive filler results in high

and varying electrical resistance[30], and the control of their actuation fields proves complex and lacks scalability.

Recent advancements in 3D printed continuous fiber-reinforced composites[31,32] offer an attractive solution to the issues when using SMP-based 4D printing to realize active origami. These composites demonstrate superior mechanical properties than composites with other reinforcement forms[33–36]. Additionally, an improved global shape-shifting ability through Joule heating[37–40] of continuous

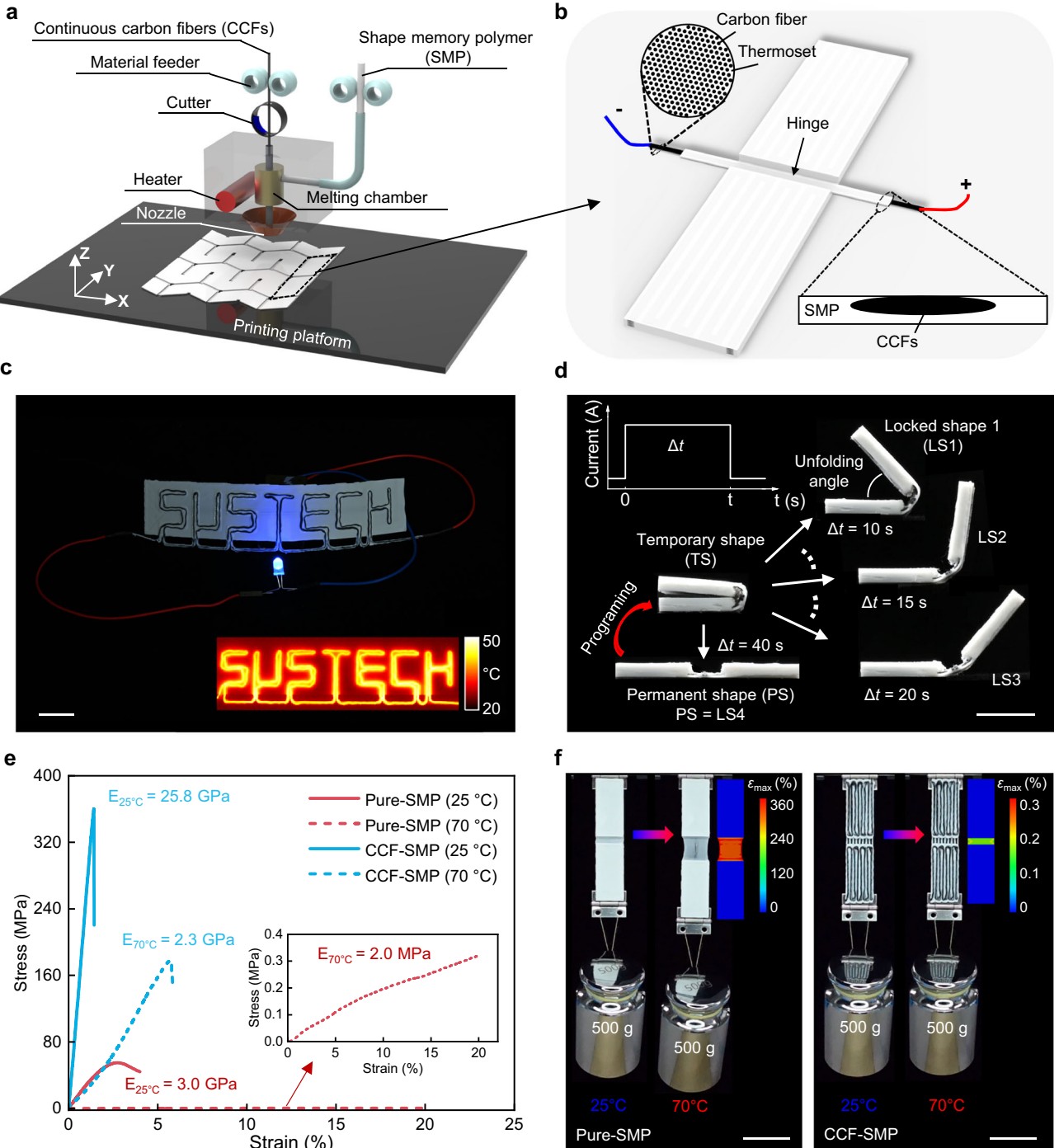

**Fig. 1 | PCEO printed by multifunctional continuous carbon fiber-reinforced SMP 3D printing technology. a** Illustration of 3D printing of CCF-SMP PCEO composite. **b** Schematics of the electrothermal hinge enabled by integrating Joule heating CCFs. **c** Demonstration of the electrical conductivity and heating capacity of CCFs. Scale bar, 20 mm. **d** Precise control of the CCF-SMP PCEO showcased by a simple hinged strip structure. Scale bar, 10 mm. **e** Stress-strain curves of the pure-SMP and CCF-SMP at 25 °C and 70 °C. **f** Demonstration of the stiffness improvement of CCF-SMP due to the incorporation of CCFs through experiment and FEA simulation. Scale bars, 20 mm.

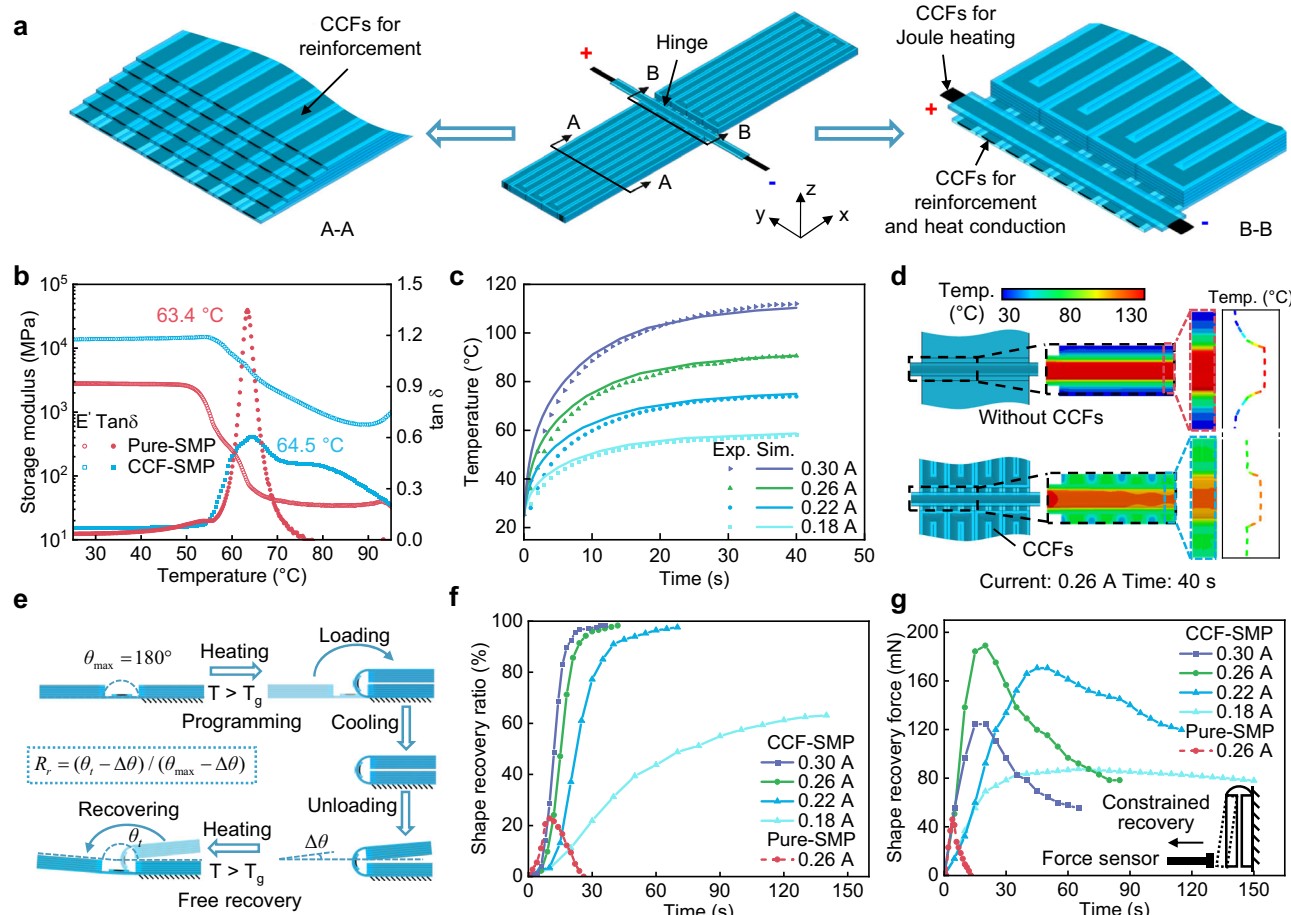

**Fig. 2 | Heating and shape recovery performance of the shape memory composite. a** Schematic illustration of the electrothermal hinged strip structure and three functions of CCFs (CCFs in the stiff panels and hinges for reinforcement, CCFs in the hinges for heat conduction, and CCFs on the hinges for localized Joule heating). **b** The DMA characterization of pure-SMP and CCF-SMP materials. **c** Influence of applied current on the heating rate of the CCF-SMP hinge. **d** Thermal field distribution of the pure-SMP (without CCFs) and CCF-SMP (with CCFs) hinges under 0.26 A Joule heating for 40 s. **e** Free recovery process and the definition of the shape recovery ratio of the electrothermal hinged strip structure. **f** Influence of applied current on the shape recovery ratio of the CCF-SMP hinge, and that of the pure-SMP hinge under 0.26 A Joule heating. **g** Influence of applied current on the shape recovery force of the CCF-SMP hinge under constrained recovery, and that of the pure-SMP hinge under 0.26 A Joule heating. Inset: Schematic showing the test method of shape recovery force.

conductive fibers in SMP matrices has been demonstrated with simple structures[37,38] or classical lightweight designs, e.g., 2D cellular structures[39,40]. However, despite this promising solution, several challenges in the 4D printing of continuous fibers-reinforced composites must be addressed to successfully create active origami with high load-bearing capacity and precise control. Firstly, the design of rigid-foldable origami necessitates a more intricate fiber layout to engineer local stiffness. Moreover, the multi-physical and highly nonlinear deploying process of these composites remains poorly understood. There is also a need to establish related modeling and control methods, which are pivotal for the precise deployment of active origami, ultimately enabling on-demand modulation of its properties for multi-scenario and multi-task applications.

Here, we report an integrated fabrication-design-actuation method to build a type of precisely controlled electrothermal origami (PCEO) with excellent mechanical performance and spatio-temporal controllability via fused deposition modeling (FDM) based 4D printing of continuous fiber-reinforced composites. We integrate continuous carbon fibers (CCFs) for Joule heating onto the hinges of the origami structure to realize controlled shape-shifting. By embedding CCFs into the PCEO, we significantly enhance the stiffness of SMP in the rubbery state, reaching 2.3 GPa, which is 1000 times greater than pure SMP. Meanwhile, the CCFs also greatly improve the uniformity of thermal distribution at the hinge facilitated by its high thermal

conductivity. Moreover, we develop an electro-thermal-mechanical model to simulate the shape memory behavior of PCEO and prove that the shape-shifting process can be precisely controlled by manipulating activation parameters, allowing the PCEO to convert into arbitrary configurations with precise positional locking and complex deploying routines. We demonstrate a broad set of applications benefiting from the capabilities of on-demand modulation for properties, i.e., geometrical and mechanical properties, of this PCEO: reconfigurable robot gripper, mechanical-tunable Miura-origami unit, mechanical-customizable architected material, and airfoil-adjustable wing. The proposed approach offers an efficient way to construct and control active origami devices and machines, broadening the practical engineering applications of origami.

## Results

### Manufacturing process and working principle

As depicted in Fig. 1a, we fabricate the PCEO using an FDM-based 3D printer designed for continuous carbon fiber-reinforced shape memory polymer (CCF-SMP) composites. The process involves melting SMP and impregnating it into CCFs, followed by coextrusion on the printing platform to create patterns of each layer according to the predetermined printing path. Figure 1b illustrates the design strategy employed to construct CCF-SMP PCEO, wherein a single CCF-SMP composite bead is deposited on the inner surface of the hinge to

provide localized Joule heating, thereby triggering shape-shifting. The zoomed-in image illustrates that the CCFs are a composite filament that has about 1500 carbon fibers pre-impregnated with thermosets. As demonstrated in Fig. 1c and Supplementary Fig. 1, we print a custom-designed conductive circuit featuring the "SUSTECH" pattern. Owing to the remarkable conductivity ($3.37 \times 10^4$ S·m$^{-1}$) of CCFs, an LED bulb connected to the circuit can be lightened up by applying 0.1 A current. Through the infrared image, we observe that the CCF circuit can quickly heat up to 50 °C within 30 s through Joule heating (Supplementary Movie 1), which generates 74.7 J of heat according to Joule's law. In Fig. 1d, due to the shape memory effect, the printed flat hinge (permanent shape) can be programmed to the fully folded shape (temporary shape) through a typical thermomechanical programming cycle for SMP[41]. By exploiting the Joule heating process of CCFs, we can precisely control the unfolding angle of the hinge by adjusting the heating time ($\Delta t$) of the applied current. This feature allows the hinge to be reconfigured and locked in an arbitrary configuration during unfolding.

Moreover, other than imparting the electrothermal effect to the printed SMP hinge, the CCFs also strongly improve the mechanical properties of SMP. As compared in Fig. 1e, the incorporation of CCFs increases the modulus of SMP from 3.0 GPa to 25.8 GPa at 25 °C, where the SMP is at its glassy state. Supplementary Fig. 2 also presents the effect of fiber arrangement parameters (such as fiber angle and spacing) on the mechanical property of CCF-SMP. More remarkably, the CCFs enhance the rubbery state modulus (tested at 70 °C) of the SMP by three orders of magnitude from 2.0 MPa to 2.3 GPa. In Fig. 1f, we conduct comparative experiments to visually demonstrate the stiffness enhancement by the incorporation of CCFs. We hang a 500 g weight onto pure-SMP and CCF-SMP hinged strips (Supplementary Fig. 3). After being heated from 25 °C to 70 °C, the pure-SMP hinge deforms by 360% to balance the weight due to the low modulus (2.0 MPa) at its rubbery state. In contrast, the deformation on the heated CCF-SMP hinge is only 0.3% as the rubbery-state modulus has been significantly increased to 2.3 GPa by the CCFs.

## Characterization of the shape memory composite

Figure 2a illustrates a simple PCEO made with CCF-SMP, where CCFs are incorporated into stiff panels and hinges for different roles. CCFs in the stiff panels and hinges are used for reinforcement, considering their excellent mechanical properties. Additionally, we utilize CCFs in the hinges for a more uniform thermal distribution due to their high thermal conductivity. Lastly, CCFs printed on the hinges provide localized Joule heating for activating the shape-shifting of origami by exploiting their conductivity.

To unravel the impact of incorporating CCFs on the thermomechanical behavior of the SMP material, we conduct dynamic mechanical analysis (DMA) tests on both pure-SMP (i.e., Polylactic acid, PLA) and CCF-SMP composite (i.e., CCF-PLA). Figure 2b shows that the storage modulus of both pure-SMP and CCF-SMP materials significantly decreases with an increase in temperature. The glass transition temperature ($T_g$) of the CCF-SMP is identified to be 64.5 °C according to the peak of tan δ (the ratio between the loss modulus and storage modulus), which is slightly higher than that (63.4 °C) of pure-SMP. More importantly, the incorporation of CCFs greatly improves the storage modulus of SMP, consistent with the results from uniaxial tensile tests shown in Fig. 1e.

The effect of the applied current on the heating rate is investigated by measuring the average temperature of the upper surface of the CCF-SMP hinge (Fig. 2c). Upon powering on, the temperature increases rapidly above the $T_g$ of CCF-SMP within 20 s when the current is greater than or equal to 0.22 A and is then gradually stabilized thereafter. The heating of CCFs is a highly nonlinear process as the resistance of CCFs changes with temperature, and the heat dissipation rate of CCFs changes with temperature and time. To further

understand the heating process, we theoretically analyze the relationship between the heat generated by Joule heating and the heat consumption and dissipation (Supplementary Note 1), that the temperature of the CCFs $T(t)$ as a function of heating time $t$ can be described as

$$T(t) = T_0 + (T_s(I, \rho_0, \alpha_F, K_F) - T_0)(1 - exp(-t/\tau_F(I, \rho_0, \alpha_F, K_F))), \quad (1)$$

where $T_0$ is the ambient temperature, $T_s$ and $\tau_F$ are the steady-state temperature and thermal time constant of fibers, which are functions of the current $I$, resistivity $\rho_0$ at $T_0$, temperature coefficient of resistance $\alpha_F$, and overall heat transfer coefficient $K_F$ of fibers. Details for $T_s$ and $\tau_F$ can be found in Supplementary Note 1, and values of $\rho_F$, $\alpha_F$, and $K_F$ are determined through experiments shown in Supplementary Fig. 4. The corresponding heat flux is calculated based on Eq.(1) and used in the simulation of the heating process (Supplementary Note 1 and Fig. 5). The simulation results closely match the experimental results (Fig. 2c). Besides, Fig. 2d shows that the integration of CCFs into the hinge also greatly improves heat conduction within the hinge area. As evident in the temperature distribution perpendicular to the fold line, the incorporation of CCFs significantly decreases the temperature gradient under 0.26 A of Joule heating for 40 s, manifesting a more uniform thermal distribution. This uniformity ensures a higher shape recovery ratio and force, as further discussed in the following two paragraphs.

To investigate the influence of the applied current on the deploying process of the proposed PCEO, we characterize the shape recovery process of the hinge with different stimulation currents. Figure 2e illustrates the definition and experimental protocol for the shape recovery ratio. First, one stiff panel is fixed while the other is free and the angle between two stiff panels is denoted as $\theta$. The SMP hinge is heated above its $T_g$ by Joule heating for 30 s and then manually deformed from the permanent shape (PS, as printed shape) $\theta_{max} = 180°$ to the temporary shape (TS) $\theta = 0°$, and then allowed to cool by powering off. Once it is cooled to room temperature, the TS is held by loading a 500 g weight for 5 minutes, then unloaded, and the programmed TS is fixed with a small bounce-back angle, $\Delta\theta$. The shape fixation ratio is defined as

$$R_f = (\theta_{max} - \Delta\theta)/\theta_{max}. \quad (2)$$

The results show that the shape fixation ratio increased with higher current levels, ranging from 88.3% at 0.18 A to 97.6% at 0.30 A (Supplementary Fig. 6). The temperature is then gradually increased for free recovery of the programmed TS via Joule heating. During the free recovery process, the angle between two stiff panels gradually increases over time, which is denoted as $\theta_t$ and serves as a metric for characterizing the extent of shape recovery. The time-dependent angle $\theta_t$ is recorded during the free recovery process and the shape recovery ratio is defined as

$$R_r = (\theta_t - \Delta\theta)/(\theta_{max} - \Delta\theta), \quad (3)$$

where the numerator represents the recovery angle of the SMP hinge over time while the denominator represents the maximum recovery angle. As shown in Fig. 2f, a higher current also resulted in a faster shape recovery process and a larger shape recovery ratio from 63.1% at 0.18 A to 98.1% at 0.30 A. Snapshots and thermal images capturing the different stages of the CCF-SMP hinge's free shape recovery are presented in Supplementary Fig. 7, and the repeatability of the CCF-SMP hinge is also demonstrated in Supplementary Fig. 8. Moreover, we conduct the same experiment to compare the shape recovery ratio of the pure-SMP hinge and CCF-SMP hinge with an applied current of 0.26 A. Figure 2f shows an incomplete recovery of the pure-SMP hinge, with a maximum recovery ratio of only 22.8%. The low recovery ratio

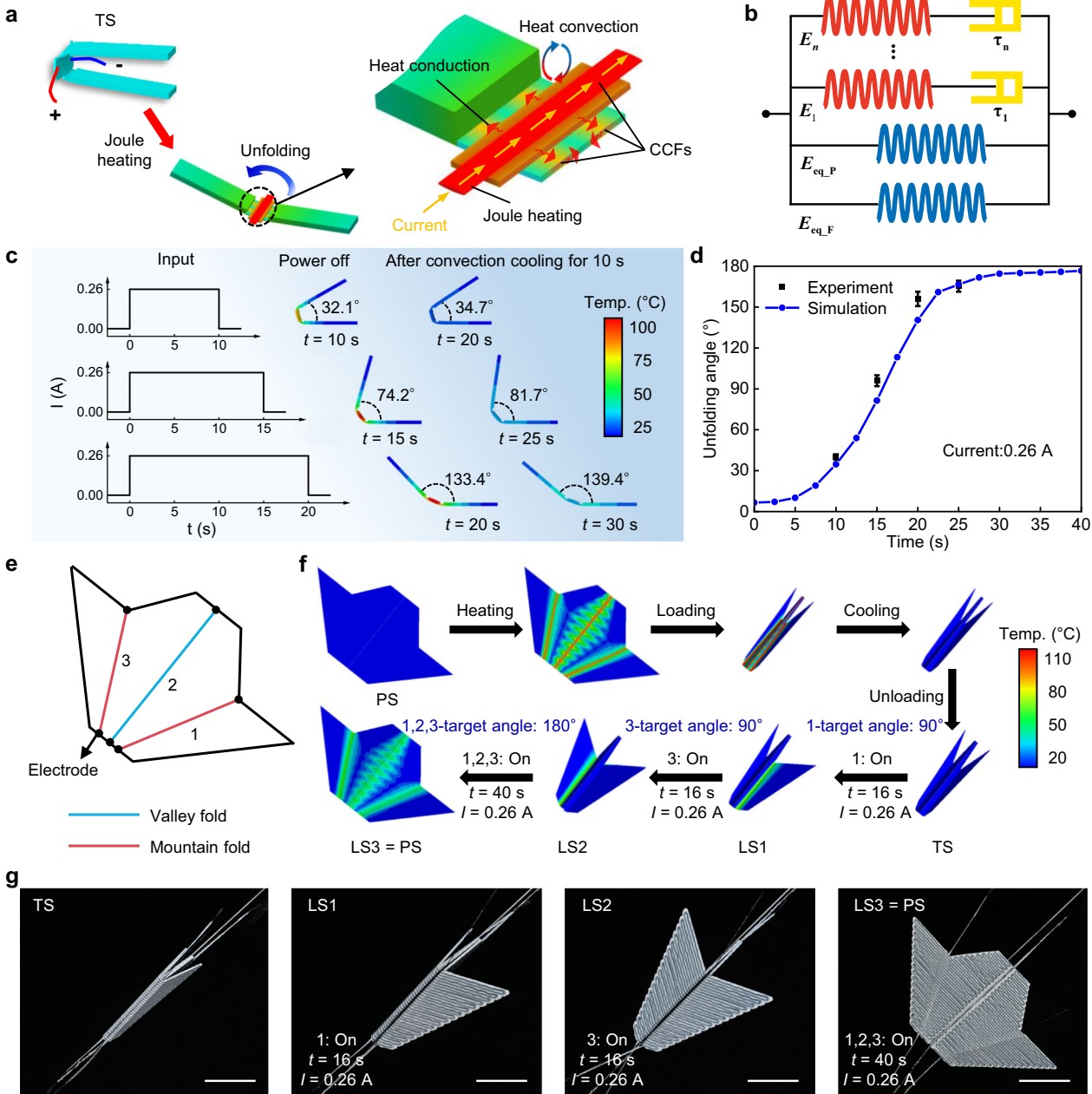

**Fig. 3 | Precisely control of electrothermal origami realized by FEA and experiment. a** The multi-physics model of the proposed PCEO. **b** Multi-branch thermoviscoelastic model. **c** Simulated temperature distribution and the unfolding angle at power off and after convection cooling for 10 s of the CCF-SMP hinge under different heating times. **d** Influence of heating time on the unfolding angle of the CCF-SMP hinge under 0.26 A Joule heating, three tests are conducted on samples for each data point, and the error bars present the standard deviation of the three repeated data. **e** The crease pattern of the airplane-shaped PCEO structure. **f** Programming of the temporary shape and actively controlled recovery of the PS of the airplane-shaped PCEO structure (LS1, LS2, and LS3 achieved by connecting both end electrodes of hinges 1, 3, and 1-2-3 at a time) realized by FEA. **g** Snapshots of actively controlled recovery of the PS of the airplane-shaped PCEO structure. Scale bars, 40 mm.

can be attributed to the coexistence of the glassy state, rubbery state, and melting state at the hinge caused by the low thermal conductivity of the SMP and resulting hot spots.

The shape recovery force of the SMP hinge is another notable factor that affects the deployment of PCEO. The inset of Fig. 2g depicts the measurement method of shape recovery force, where one end of the SMP hinge is fixed while the other end is in contact with a force sensor. The force sensor tracks changes in force values as the strained TS attempts to recover its PS after being stimulated by Joule heating (Supplementary Fig. 9). It should be noted that the SMP hinge is positioned perpendicular to the ground to minimize the influence of gravity on the measurement of shape recovery

force. As depicted in Fig. 2g, the recovery force shows a trend of first increasing and then decreasing over time. Moreover, the maximum recovery force of the CCF-SMP hinge experiences variation with the increase of applied current. It reaches the peak at 0.26 A when most of the CCF-SMP hinge is in the rubbery state. To underscore the pivotal role of CCFs, we compare the maximum shape recovery force of both pure-SMP and CCF-SMP hinges under 0.26 A Joule heating, as depicted in Fig. 2g. The maximum shape recovery force of the pure-SMP hinge is only 46.1 mN, significantly smaller than that of the CCF-SMP hinge, which records 188.9 mN. The recovery force of the pure-SMP hinge is too small to overcome gravity and recover to its permanent shape (Supplementary Fig. 10). The shape

recovery process of both the pure-SMP and CCF-SMP hinges can be seen in Supplementary Movie 2.

### Precise control of the electrothermal origami

To achieve precise control of the electrothermal origami, we develop a multi-physics simulation model about the deploying process by finite element analysis with the ABAQUS software. The finite element model for PCEO considers electrical, thermal, and mechanical effects for its deployment control process. Figure 3a depicts the electro-thermal processes considered in the simulation, including Joule heating of CCFs, heat conduction in hinges and stiff panels, and heat convection between air and hinges and stiff panels. For the electrothermal origami, the increment of internal energy in the transient state is balanced by the heat generated by Joule heating, the heat developed by conduction, and the heat loss through convection. Let us consider the amount of heat evolved per unit time and volume in the electrothermal origami, the generalized governing differential equation for the temperature distribution is given by

$$\rho c \frac{\partial T}{\partial t} = \dot{\Phi}_{JH}(t) + k\nabla^2 T - \dot{\Phi}_{CONV}. \tag{4}$$

The left term is the increment of internal energy in the transient state, where $\rho$ and $c$ are the density and specific heat of the composite, respectively. $\dot{\Phi}_{JH}(t)$ is the generated Joule heat per unit time and unit volume of CCFs, which follows as

$$\dot{\Phi}_{JH}(t) = \frac{I^2\rho_0}{S^2}(1 + \alpha_F(T(t) - T_0)), \tag{5}$$

where $S$ is the cross-sectional area of CCFs. The second term on the right pertains to ordinary heat conduction in hinges and stiff panels, where $k$ is the thermal conductivity of the composite. $\dot{\Phi}_{CONV}$ is the natural convection we considered in the heat transfer analysis to capture the true thermal response of the electrothermal origami. The heat loss through convection per unit time and unit volume is expressed as

$$\dot{\Phi}_{CONV} = \frac{hA}{V}(T_s - T_a), \tag{6}$$

where $h$ is the convection coefficient (More details about the determination of $h$ are presented in Supplementary Fig. 11), $A$ and $V$ are respectively the convection surface and volume, $T_s$ is the convection surface temperature, and $T_a$ is the air temperature near the convection surfaces. It should be noted that considering heat loss through convection is crucial for the heat transfer analysis of the electrothermal origami (Supplementary Fig. 12).

More importantly, we establish a multi-branch thermoviscoelastic model (Fig. 3b) to describe the thermo-mechanical response of CCF-SMP used in the PCEO. The model consists of two equilibrium branches of SMP and CCFs and several thermoviscoelastic nonequilibrium branches (number of $n$) to present the multiple relaxation processes of the polymer. By applying the Boltzmann's superposition principle, the total stress $\sigma$ as a function of time $t$ is expressed as

$$\sigma(t) = (E_{eq\_F} + E_{eq\_P})e(t) + \sum_{i=1}^{n} E_i \int_0^t \frac{\partial e(s)}{\partial s} \exp\left[-\int_s^t \frac{dt'}{\tau_i(T)}\right] ds, \tag{7}$$

where $e$ is the total strain, $E_{eq\_P}$ and $E_{eq\_F}$ are the elastic modulus of equilibrium branches of polymers and fibers, and $E_i$ and $\tau_i(T)$ are respectively the elastic modulus and the temperature-dependent relaxation time in the $i$th nonequilibrium branch. After determining the model parameters by using the DMA results of CCF-SMP (see Supplementary Note 2, Supplementary Fig. 14, and Supplementary

Table 1)[42,43], the model shows the capability of capturing the temperature-dependent free recovery behavior of PCEO under Joule heating (Supplementary Fig. 15). A high degree of consistency between simulation and experiment data (Supplementary Fig. 16). Based on this validated model, we find that the PCEO can be locked in an intermediate state during the free recovery process by controlling the duration of the electrical current input (Fig. 3c). This phenomenon is facilitated by the unique attributes of CCFs, which can rapidly reach high temperatures upon activation and cool down quickly upon deactivation. Figure 3c presents the PCEO undergoing natural convection cooling after deactivation, resulting in a significant slow-down of the recovery process. Eventually, the PCEO reaches room temperature within approximately 10 s and its shape is locked in an intermediate state. The unfolding angle of PCEO is mainly governed by the heating time when the applied current is fixed. Here, a stimulation current of 0.26 A is employed, as it provides the maximum shape recovery force and fast shape recovery. Figure 3d shows the unfolding angle with different heating times predicted by the simulation. We conduct experiments with four heating times to validate the model and confirm its effectiveness in guiding the precise control of the shape-shifting of the PCEO. More details about the establishment and guidance of the multi-physics model for PCEO are presented in Supplementary Fig. 17, and the procedure of simulation can be seen in Supplementary Fig. 18.

The electrothermal activation and deactivation of the PCEO allow shape-shifting with spatiotemporal control, including localized control, unfolding angle control, and sequence control. To demonstrate this capability, we utilize an airplane-shaped PCEO structure as an example. Figure 3e shows the crease pattern of the airplane-shaped PCEO, consisting of one valley- (blue) and two mountain- (red) creases, with electrodes located at both ends (Supplementary Fig. 19). Figure 3f illustrates a stepwise actively controlled recovery process with a complex shape-shifting through multi-physics modeling. The process includes seven steps: i) applying power to all hinges to heat them above their $T_g$ via the Joule effect; ii) programming the airplane-shaped PCEO into its fully folded shape; iii) cooling the hinge temperature to room temperature by powering off; iv) unloading the structure, causing it to transform into its TS with a slight bounce-back; v) selectively connecting Hinge 1 to the power source with 0.26 A for 16 s (as guided by Fig. 3d) to achieve a 90° deployment of the right-wing target angle (LS1); vi) selectively connecting Hinge 3 to the power source with 0.26 A for 16 s to achieve a 90° deployment of the left-wing target angle (LS2); vii) connecting all hinges to the power source with 0.26 A for 40 s to recover the structure to its PS (LS3) for reprogramming purposes. Experiment snapshots (Fig. 3g) and a movie (Supplementary Movie 3) further corroborate the effectiveness of the multi-physics model. This approach allows for selective and localized Joule heating to trigger the precise deployment of specific hinges at different times without coupling with others.

### Reconfigurability of the PCEO

The shape reconfigurability is needed for active origami to satisfy multi-scenario and multi-task applications. Here, we present a shape-reconfigurable PCEO strip structure with spatiotemporal controllability via manipulating the region and time of Joule heating. Figure 4a depicts the crease pattern of the reconfigurable PCEO strip structure (Supplementary Fig. 20) consisting of seven hinges arranged alternately between valley and mountain creases (numbered 1-7 from top to bottom). Programing PS to TS in the aforementioned way endows the PCEO with high packaging efficiency. By applying different inputs guided by the multi-physical simulation, the PCEO strip structure can be deployed into multiple configurations, including single-floor, double-floor, and metamaterials unit configurations (Fig. 4a). An automatic control and deployment process of PCEO is exemplified and illustrated in Supplementary Fig. 21 and Supplementary Movie 4. Additionally, control of the actuation sequence for the PCEO strip

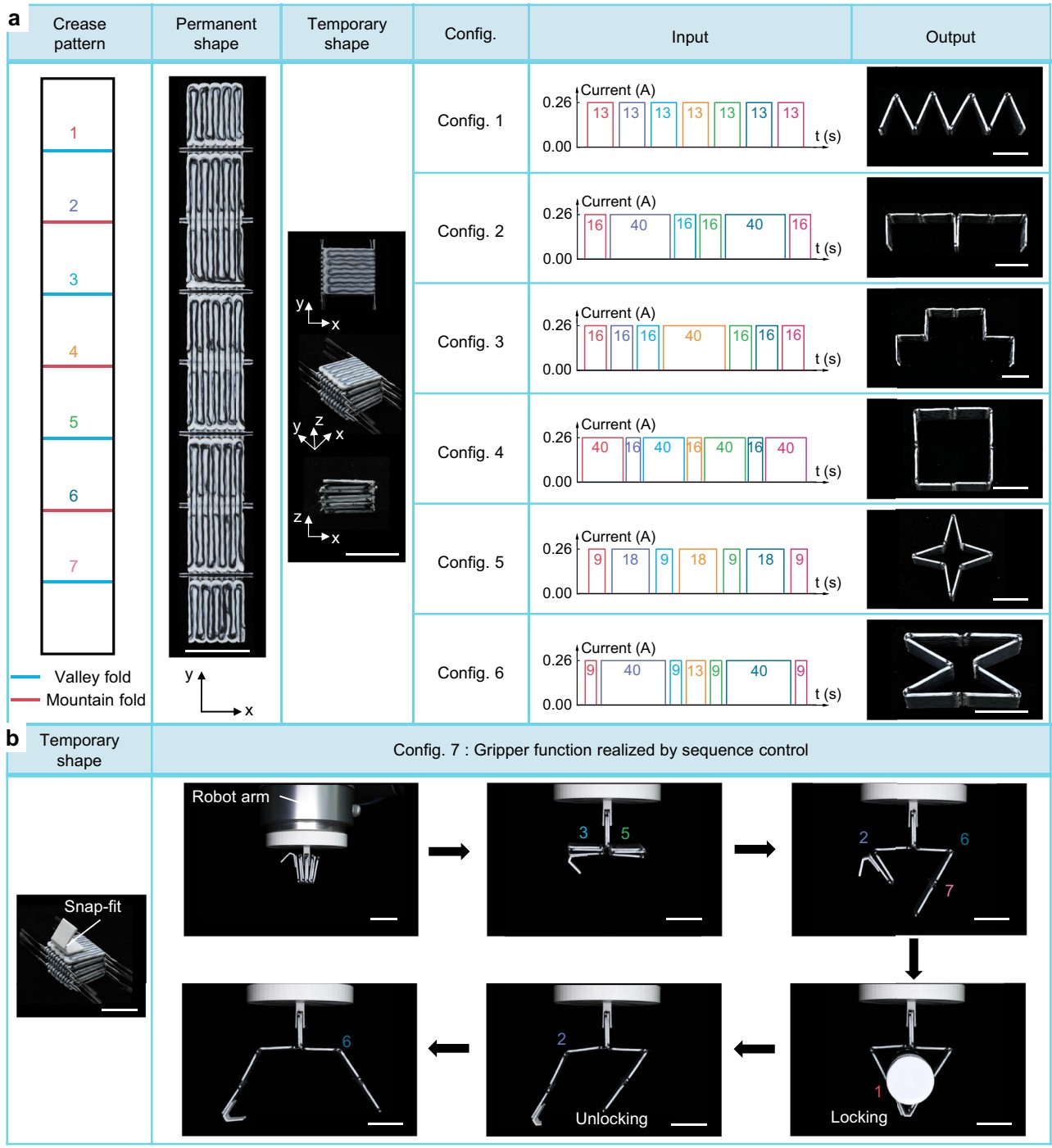

**Fig. 4 | Reconfigurable PCEO. a** The crease pattern and snapshots (the permanent and temporary shapes) of the reconfigurable PCEO strip structure. Hinges are numbered 1 to 7 from top to bottom. The input current and heating time (as guided by simulation) of each hinge and experimental snapshots of different configurations. Scale bars, 20 mm. **b** Demonstration of the gripper functional configuration by sequence control. Snapshots of the temporary shape and the whole working process (grasp and release a solid adhesive rod) of the robotic gripper. A 3D-printed snap-fit is attached to the end of the PCEO strip structure to achieve self-locking in the working process. Scale bars, 20 mm.

structure can provide additional functionalities, i.e., grasping, as showcased in Fig. 4b. A robotic gripper is created by assembling the PCEO strip onto a 3D-printed base, which is attached to an industrial robotic arm (UR10, Universal Robots, Denmark) for vertical motion. A 3D-printed snap-fit mechanism is attached to the end of the strip to achieve self-locking during the deployment process. The self-locking and unlocking capacity of the robotic gripper is demonstrated in Fig. 4b, where the gripper can grasp and release a solid adhesive rod by precisely designing the actuation sequence and unfolding angle of

each hinge (Supplementary Movie 5). It should be noted that all configurations originate from one PCEO strip structure, and its reconfigurable deployment requires reprogramming.

## Tunable mechanical properties of precisely controlled electrothermal Miura-origami

The reconfigurability of the PCEO enables the design of a precisely controlled electrothermal Miura-origami (PCEM) unit with tunable mechanical properties by manipulating activation parameters.

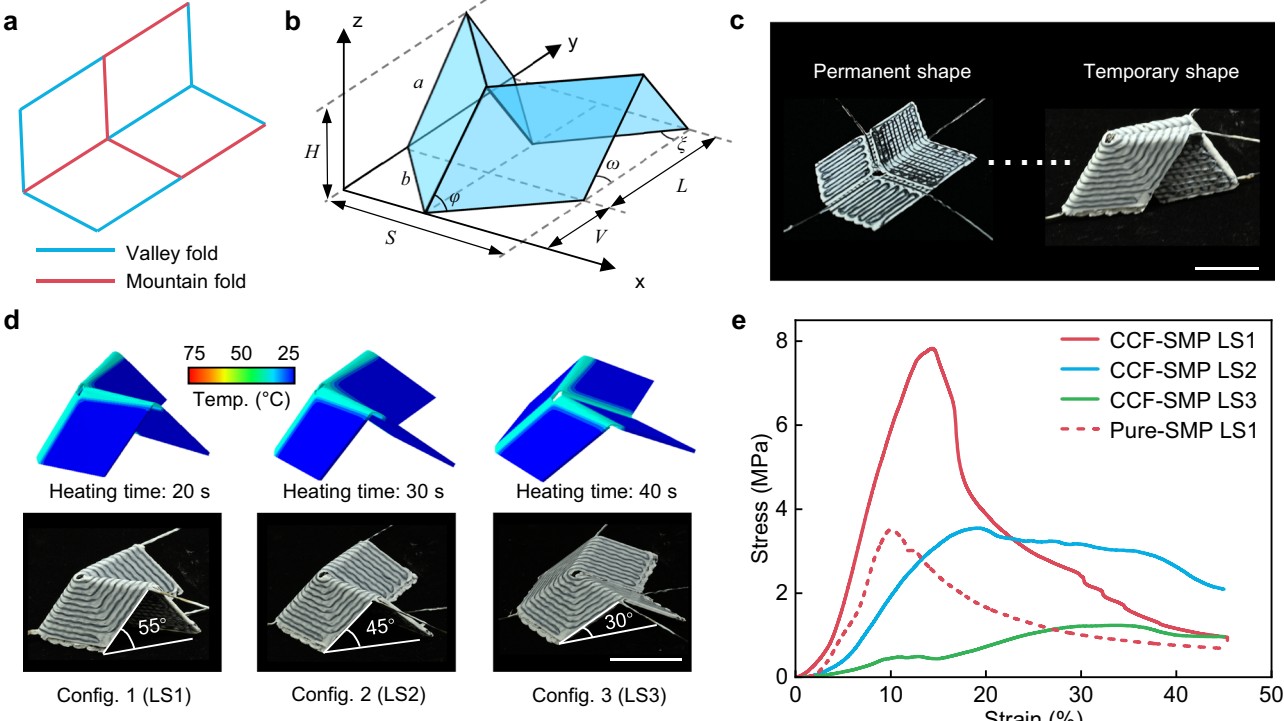

**Fig. 5 | PCEM with tunable mechanical properties. a** The crease pattern of the Miura-origami unit. **b** A folded Miura-origami sheet consists of tessellations of a unit. The unit geometry can be described using a parallelogram with sides $a$, $b$, and acute angle $\varphi$, along with the deployment angle $\omega \in [0, \pi]$. An alternative parameterization is given by dimensions $H$, $S$, $V$, and $L$. **c** Snapshots (the permanent and temporary shapes) of the PCEM. Scale bar, 20 mm. **d** Experimental and simulation studies on different configurations of the PCEM activated by different heating times under 0.26 A stimulus. Scale bar, 20 mm. **e** Compression tests on the pure-SMP and CCF-SMP folded Miura-origami thick panels with different deployment angles.

Figure 5a shows the crease pattern of a Miura-origami unit, where the mountain and valley creases are alternately arranged. When folded into a 3D structure, the geometry of the unit can be parameterized in several ways[44]. Here, the unit is defined by the dimensions of its smallest constituent component, a parallelogram with sides $a$, $b$, and acute angle $\varphi$, along with the deployment angle $\omega \in [0, \pi]$ (Fig. 5b). The geometric relationships of the Miura-origami unit are presented in Supplementary Note 3. As depicted in Fig. 5c, the 4D printed PCEM unit is first programmed from PS to TS manually after heating all hinges to a temperature above their $T_g$ via Joule heating. Subsequently, the strained TS is fixed by rapid cooling to room temperature. Figure 5d shows three configurations with different designed deployment angles (55°, 45°, and 30°) that can be deployed by changing the heating time under the guidance of the multi-physics model (20, 30, and 40 s for LS1, LS2, and LS3, respectively). The simulation demonstrates that the PCEM can quickly cool down below $T_g$ within 10 s upon deactivation. The deployment processes are presented in Supplementary Movie 6. Adding a stopper under different deployed structures can transform them into load-bearing structures (Supplementary Fig. 22b). The geometric design parameters are listed in Supplementary Table 2. The stopper can be assembled with the deployed structure through physical locking (Supplementary Fig. 23). In compression tests on the Miura-origami with different deployment angles, both the compressive modulus and strength progressively increase as the deployment angle increases from 30° to 55° (Fig. 5e). Specifically, the stiffness and strength can be varied by an order of magnitude, forming a large performance space. However, in comparison, the compressive strength of the pure-SMP Miura-origami is less than half that of CCF-SMP Miura-origami with the same deployment angle (55°).

## Combinatory digital Miura-origami architected materials

The programmability of architected materials refers to the ability to achieve the on-demand modulation of mechanical property by producing nonuniformly distributed regions of deformation[45–48]. Here, we propose a combinatory digital architected material based on PCEM units, whose mechanical properties can be modulated by simultaneously considering geometric design and deployment control. Figure 6a illustrates that the target three-dimensional Miura-origami structures with different deployment angles ($\omega_i$) and heights ($H_i$), which can be designed by varying geometric ($a_i$ and $\varphi_i$) and activation parameters (current $I_i$ and heating time $t_i$) based on geometric relationships (Supplementary Note 4 and Fig. 24) and the electro-thermo-mechanical finite element model. These deployed Miura-origami structures can be plugged into a bottom plate with a periodically arranged stopper array in arbitrary combinations, effectively decoupling the deformation constraints between interconnected building blocks and greatly increasing the deformability and programmability of materials. The mechanical properties of the combinatory digital architected material can be obtained by linearly superposing the compressive force-displacement curves of these decoupled units. Here, we take Miura-origami with three different deployment angles of $\omega_1 = 50°$, $\omega_2 = 45°$, and $\omega_3 = 35°$ as the basic unit (named $Ori_1$, $Ori_2$, and $Ori_3$, respectively) for linear superposition. The geometric design and activation parameters for Miura-origami units with different deployment angles and the geometric design for the bottom plate are listed in Supplementary Table 3. Figure 6b shows the compressive force-displacement curves of different basic units, which have different starting points due to their varying heights. We fabricate a 3 × 3 array architected material prototype based on $Ori_1$, $Ori_2$, and $Ori_3$ units to validate the feasibility of customizing the combinatory digital architected materials, which can be further expanded in X, Y, and Z

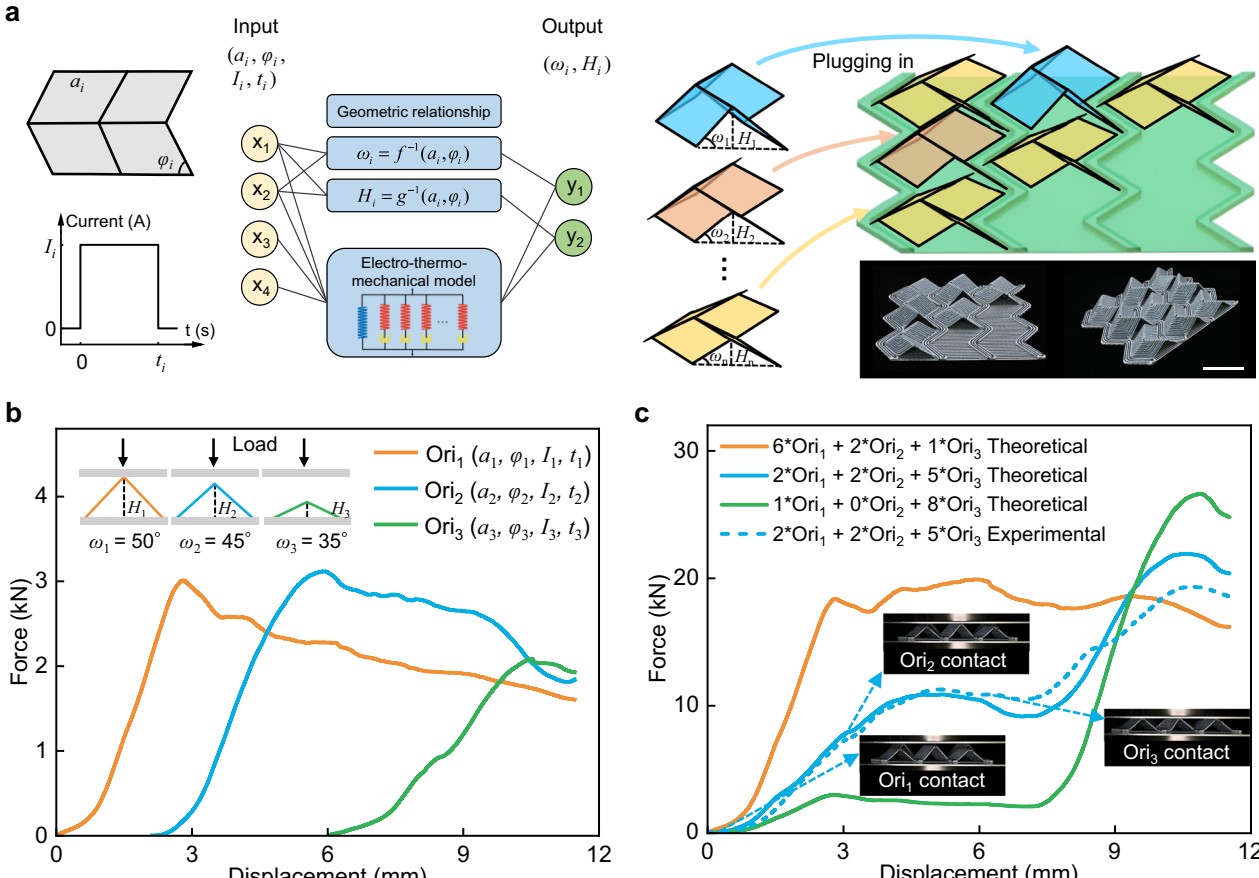

**Fig. 6 | Programmable mechanical response of an array of Miura-origami structures. a** Design and fabrication of the combinatory digital architected material consisting of tessellations of $3 \times 3$ array units. Scale bar, 40 mm. **b** Force-displacement curves of three-dimensional Miura-origami structures (named $Ori_i$, $i = 1, 2, 3$) with different deployment angles. Inset: Schematic showing the test method of compression. **c** Typical target force-displacement curves (obtained by linearly superposing the compressive force-displacement curves of basic units) with different combination forms and the experimental result of combination form of two $Ori_1$s, two $Ori_2$s, and five $Ori_3$s. Inset: Snapshots when the load cell contacts different Miura-origami units.

directions (Supplementary Fig. 25). The deployment, assembly, and compression process is presented in Supplementary Movie 7. Figure 6c shows typical target force-displacement curves obtained by different combinations of basic units, such as multi-stage and quasi-zero stiffness. We experiment with a combination form of two $Ori_1$s, two $Ori_2$s, and five $Ori_3$s to further demonstrate the programmable effectiveness of these combinatory digital architected materials. As shown in Fig. 6c, by exploiting the different heights of different basic units, $Ori_1$, $Ori_2$, and $Ori_3$ are sequentially contacted and compressed as the compression proceeds to modulate the mechanical behavior of the architected material.

**Variable thickness wing based on PCEM**
The spatiotemporal controllability of the PCEM allows for the design of a variable thickness wing with reconfigurable geometry and inversely designable airfoil for multiple flight scenarios by combining the deployable Miura-origami units with different geometry designs. To demonstrate the geometry variability, we showcase a precisely controlled electrothermal variable thickness wing origami as an example. Figure 7a displays the crease pattern of the variable thickness origami structure consisting of five Miura-origami units with distinct geometrical designs, where $a_i$ determines the geometry of each unit (Supplementary Fig. 26). Figure 7b depicts the snapshots of the permanent and temporary shapes programming through Joule heating. As illustrated in Fig. 7c, the mode of the variable thickness wing, with a fixed geometric design, can be altered by varying the heating time, which

also demonstrates that the proposed electrothermal control method can be extended to complex-shaped origami. Additionally, different airfoils can be obtained by changing the combination of geometric parameter $a_i$ for each mode (Fig. 7d). The geometric design parameters of four different structures (I, II, III, and IV) are listed in Supplementary Table 4. Assuming that the relative position of a peak point in the horizontal and vertical coordinates represents an airfoil (Fig. 7e), Fig. 7f illustrates the design space of airfoils can be obtained by calculating the combination of all geometric parameters $\{a_1, a_2, ..., a_5\}$ for each mode (design details can be seen in Materials and Methods). This design space serves as a guide for the inverse design of the airfoils for the variable thickness wing. Nevertheless, a more challenging task to be explored is the inverse design of the entire cross-sectional profile of the airfoil, rather than focusing solely on its relative peak point position. This requires a more advanced computational-based method. Also, the precise deployment control of the skin is often indispensable in practical applications to conform to the airfoil of the wing, enabling both large deformation and shape-locking.

## Discussion
We report a simple yet versatile fabrication-design-actuation methodology for precisely controlled electrothermal origami with excellent mechanical performance and spatiotemporal controllability. This method involves the integrated fabrication of PCEO via FDM-based 4D printing for continuous fiber-reinforced composites by depositing CCFs for Joule heating onto the hinges of the origami, which enables it to

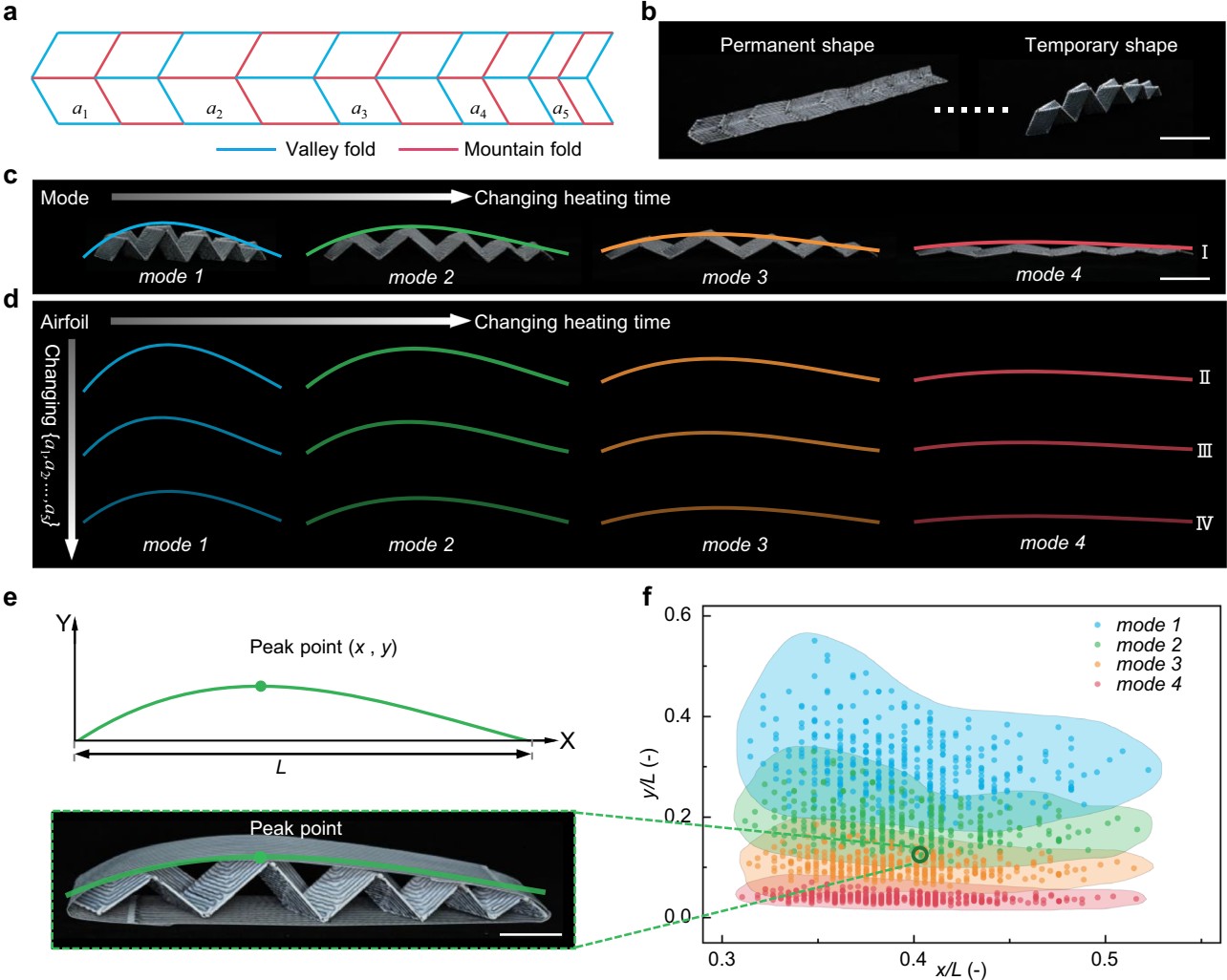

**Fig. 7 | Design of variable thickness wing. a** The crease pattern of a variable thickness wing consists of tessellations of five Miura-origami units. The unit geometry can be designed by changing side *a*. **b** Snapshots (the permanent and temporary shapes) of the variable thickness wing. Scale bar, 40 mm. **c** Experimental demonstration of different modes of the variable thickness wing obtained by changing heating time. Scale bar, 40 mm. **d** Variation of airfoils of the variable thickness wing with different geometric designs. **e** One airfoil can be represented by the peak point coordinates (*L* is the wing extension). Scale bar, 20 mm. **f** Design space of the variable thickness wing.

morph by electrical actuation. Through embedding CCFs into a thermoplastic SMP matrix, we achieve a 1000-fold increase in stiffness in the rubbery state while also greatly improving the uniformity of thermal distribution at the hinge. To guide the spatiotemporal control of the PCEO, we develop an electro-thermal-mechanical model that describes its multi-physical and highly nonlinear deploying process. We prove that the shape recovery process can be controlled by manipulating activation parameters, allowing the PCEO to achieve programmable morphing without the need for prefabricated structures and sustain different shapes without energy consumption. Furthermore, we demonstrate several applications taking advantage of the capabilities of on-demand modulation for properties, i.e., geometrical and mechanical properties, of this PCEO: reconfigurable robot gripper, mechanical-tunable Miura-origami unit, mechanical-customizable architected material, and airfoil-programmable wing. These capabilities are particularly valuable in 3D complex, lightweight, and load-bearing structures, such as disaster relief structures, actively controlled aerodynamic surfaces, deployable solar arrays, and antennas. In these multi-scenario and multi-task applications, key performance such as mechanical properties, morphing sequences, and morphing angles can be adjusted on-demand to adapt to varying requirements.

However, considerable efforts are needed to translate these design concepts, e.g., the reconfigurable robot gripper and variable thickness wing, into real-world applications. For instance, reversible deployment control can be realized by the hybrid-actuation solution including cable driven, etc., as an alternative to the bulky actuation units required by the non-responsive material system. Moreover, precise control in complex environments, e.g., dynamic loads and extreme temperatures, deserves further investigation to align with specific functional requirements.

## Methods

### Materials

The SMP material used was polylactic acid (PLA) filament (PolyLite PLA, Polymaker, China) with a diameter of 1.75 mm. The support material polyamide (PA) filament (PolyMide CoPA, Polymaker, China) was used for the printing of PCEO's hanging mountain hinge, which is easy to peel off after printing (Supplementary Fig. 27). The PLA and PA filaments were dried at 60 °C and 80 °C respectively for 10 hours before 3D printing. The fiber used was composite carbon fiber filament (CCF 1.5k, Anisoprint, Russia) with a diameter of 0.35 mm, and its fiber volume fraction is approximately 60%.

## Fabrication of the PCEO

All PCEO structures were fabricated using a commercial open-source dual-nozzle material extrusion composite 3D printer (A4, Anisoprint, Russia). The build volume of the machine is 297 mm × 210 mm × 147 mm. The left nozzle, with a diameter of 0.4 mm, supplies thermoplastics only. The right nozzle, with a diameter of 1 mm, feeds CCFs impregnated with thermoplastics (Fig. 1a). There is a cutter above the nozzle to cut the fiber during printing. Here, the default print parameters were set for all samples as follows: layer thickness of 0.2 mm, the printing speed of the left nozzle of 50 mm/s, the printing speed of the right nozzle of 9 mm/s, nozzle temperature of 210 °C, and substrate temperature of 45 °C. All printing execution files of CCF-SMP PCEO were generated by in-house program codes developed with Matlab software version R2020a (Math Works, USA).

## Temperature characterization

A thermal imager (Co. MAG 32, Magnity Technologies, China) was used to record the temperature distribution after the customized conductive circuit (Fig. 1c) was connected to a 30 V output power supply (GPD-4303S, GWinstek, China). The thermal imager was also utilized to measure the average temperature of the upper surface of pure-SMP and CCF-SMP hinges (dimensions: $3 \times 14.4 \, mm^2$) with different currents applied.

## Tensile tests of pure-SMP and CCF-SMP

Uniaxial tensile tests of pure-SMP and CCF-SMP materials were performed using a uniaxial testing machine (Criterion Model 43, MN, USA) equipped with a 10 kN load cell and a thermal chamber. Tensile strength and modulus were measured according to the standard of ASTM D638 on 3D printed SMP specimens with a size of type IV, both at room temperature (25 °C) and above the glass transition temperature (70 °C). The tensile test of 3D printed CCF-SMP specimens with a size of 250 mm × 15 mm × 1 mm was conducted according to the standard of ASTM D3039 at 25 °C and 70 °C. The loading rate of all tensile tests was 2 mm/min. In both cases, Young's modulus was determined as the initial slope of the material stress-strain curve in the linear regime. Five specimens were tested for each kind of test to obtain an average value of the targeted properties.

## Finite element simulation

The commercial FE software ABAQUS (version 6.14, Dassault Systemes Simulia Corp., USA) was used to numerically investigate the deformation between pure-SMP and CCF-SMP. The Abaqus/Implicit solver was employed for all the simulations. The hinged strip was modeled using a four-node bilinear plane stress quadrilateral element (ABAQUS element type CPS4R) with a mesh size of 0.75 mm. The material properties used for pure-SMP are according to ref. 49, and the material properties for CCF-SMP are listed in Supplementary Table 1. The input load for both hinged strips is 0.5 N, and the corresponding deformations are shown in Fig. 1f.

To further investigate the effect of applied current on the heating rate and the effect of the incorporation of CCFs on the thermal distribution of pure-SMP and CCF-SMP hinges, finite element simulations were conducted using the ABAQUS software version 2021 (Dassault Systemes Simulia Corp., USA). In the simulations, the thermal conductivity, specific heat, and density were set to 7 W m$^{-1}$ K$^{-1}$ [50], 1950 J Kg$^{-1}$ K$^{-1}$, and 1.51 g cm$^{-3}$, respectively, for CCFs and 0.13 W m$^{-1}$ K$^{-1}$ [51], 1800 J Kg$^{-1}$ K$^{-1}$ [51], and 1.24 g cm$^{-3}$, respectively, for SMP. The structure was modeled using an eight-node linear heat transfer brick element (ABAQUS element type DC3D8) for CCF-SMP and a ten-node bilinear quadratic heat transfer tetrahedron element (ABAQUS element type DC3D10) for SMP.

## Dynamic mechanical analysis (DMA) test of pure-SMP and CCF-SMP

A dynamic mechanical analysis (DMA) test was conducted on 3D-printed pure-SMP and CCF-SMP strips (30 mm × 10 mm × 1 mm) to measure their thermomechanical properties. A DMA machine ((DMA 242 E Artemis, Netzsch Instrument Inc., Germany) with a three-point bending clamp was used to perform the test. The samples were tested in a temperature ramp mode over a temperature range of 25–100 °C at a heating rate of 2 °C min$^{-1}$ and a frequency of 1 Hz.

## Shape recovery ratio measurement of pure-SMP and CCF-SMP hinge

One end of the pure-SMP or CCF-SMP hinge is horizontally fastened using a clamp, while the other end remains free. When the current is applied to the CCFs for Joule heating, the pure-SMP or CCF-SMP hinge exhibits shape recovery behavior. The shape-shifting videos were obtained using the digital camera (Nikon, Z7II) and analyzed by using Adobe After Effects 2022 software. The black background of movies and photographs is just a stylistic choice. The surface temperature of the pure-SMP or CCF-SMP hinge during morphing was measured by a thermal imager (Co. MAG 32, Magnity Technologies, China). Three specimens were tested for each kind of test to obtain an average value of the targeted properties.

## Shape recovery force measurement of pure-SMP and CCF-SMP hinge

Initially, the PCEO was powered on for 30 s, causing the programming of the permanent shape (PS) to the temporary shape (TS). Subsequently, the power was turned off, allowing the structure to cool down. Once it's cooled down to room temperature, the TS was held by loading a 500 g weight for 5 min, after which the strained TS was fixed. To measure the shape recovery force, one end of the pure-SMP or CCF-SMP hinge was positioned perpendicularly to a fixed surface, while the other end was connected to a force sensor (LS30, Lijing, China) (Supplementary Fig. 9). By stimulating the strained TS, the shape recovery force was recorded. The processes involved the TS attempting to recover its PS. The resulting data allowed for the construction of curves depicting the shape recovery force as a function of time, as shown in Fig. 2f. Three specimens were tested for each kind of test to obtain an average value of the targeted properties.

## Compressions test of the Miura-origami unit

Uniaxial compression test of 3D printed Miura-origami unit structures with limit bottom plate was performed using a 20 kN universal material testing machine (Table-top machines Z020 of the Allround-Line, ZwickRoell, Germany) at room temperature with a load rate of 4 mm/min. Three specimens were tested for each kind of test to obtain an average value of the targeted properties.

## Compression test of combinatory digital architected materials

Uniaxial compression test of combinatory digital architected material with a 3 × 3 array stopper bottom plate was conducted using a 250 kN universal material testing machine (Table-top machines Z250 of the Allround-Line, ZwickRoell, Germany) at room temperature by loading at a rate of 4 mm/min.

## Design of variable thickness wing

The design space of airfoils that was obtained by calculating the combination of all geometric parameters $a_i$ satisfies design constraints, $a_2 > a_3 > a_4 > a_5$ and $a_1 + a_2 + a_3 + a_4 + a_5 = 120$ mm. The airfoil is fitted by a third-order polynomial.

## Data availability

The data generated in this study are provided in the main article, Supplementary Information and Source data file. Source data are provided with this paper.

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

## Acknowledgements
The authors acknowledge the support from the National Natural Science Foundation of China under the grant 52105261, Guangdong Basic and Applied Basic Research Foundation (2022A1515010316), and Shenzhen Science and Technology Innovation Commission under the grant JCYJ20210324104610028. The authors acknowledge the assistance of SUSTech Core Research Facilities. Q.G. acknowledges the support from the Key Talent Recruitment Program of Guangdong Province (2019QN01Z438).

## Author contributions
Y.W., H.Y., Q.G., and Y.X. conceived the ideas and designed the research. Y.W. designed and printed all structures. Y.W. and H.Y. conducted the experiments. Y.W., H.Y., J.H., Q.G., and Y.X. analyzed the experimental results. Y.W. and H.Y. performed the analytic calculation and FEA simulation. Y.W. and Y.X. drafted the manuscript. Y.W., H.Y., Q.G., and Y.X. revised the manuscript.

## Competing interests
The authors declare no competing interests.
