## [Peer Review File · Nature Communications]

Electrothermally controlled origami fabricated by 4D printing of continuous fiber-reinforced compositesREVIEWER COMMENTS

Reviewer #1 (Remarks to the Author):

This paper introduces a method to build electro-thermal origami with continuous carbon fiber (CCF) and shape memory polymer (SMP). The paper shows fabrication method, simulation for precise control, and application scenarios of the proposed technology. The major claimed contributions include: (1) use of CCF to improve mechanical characteristics of SMP, (2) develop simulation for precise control, (3) demonstrate application potentials. Although the developments are interesting, the three aspects are not yet well-developed and convincing enough for a top-tier journal like Nature Communications. Please consider addressing the following comments before the publication can be accepted:

Fabrication of CCF within SMP:

* Embedding continuous carbon fiber (or other continuous fiber) in SMP (or other polymer) composite is not a brand-new idea. Instead, it was demonstrated in previous research [R1-R3]. For example, in the paper by Wang et. al. [R1], they have already demonstrated the idea of combining continuous fiber and 4D printing to build multi-functional systems. Applying a well-developed fabrication to build origami may not be itself an innovative enough idea for Nature Communications. Is there improvement in the techniques? Or is there other worth mentioning achievement? Please consider providing more discussion regarding the fabrication.

[R1] Qingrui Wang, Xiaoyong Tian, Lan Huang, et. al., 2018, Programmable morphing composites with embedded continuous fibers by 4D printing. *Materials and Design*, 155, 404–413.

[R3] Ryosuke Matsuzaki, Masahito Ueda, Masaki Namiki, et. al., 2016, Three-dimensional printing of continuous-fiber composites by in-nozzle impregnation, *Scientific Report*, 6:23058.

[R3] Chengjun Zeng, Liwu Liu, Wenfeng Bian, et. al., 2020, 4D printed electro-induced continuous carbon fiber reinforced shape memory polymer composites with excellent bending resistance, *Composites Part B*, 194, 108034.

* Following the previous comment, it is necessary to present a more detailed literature review regarding how prior research uses 3D/4D printing together with continuous fibers. This will help readers to identify how this work is different from prior research.

Simulation of 4D printing of origami for precise control:

* It seems that the author is assuming no conduction between the folding crease and the surrounding atmosphere as well as the panels (as forms of heat loss). However, for small-scale electro-thermal systems, conduction is usually the major form of heat loss [R4, R5]. For the scale of the reported electro-thermal origami system, it is likely that conduction is still contributing to a considerable part of the heat loss. Can the authors justify their assumption by including conduction in the model and show that conduction is indeed not the major contributor?

[R4] Hussein Hussein, Aref Tahhan, Patrice Le Moal, et. al., 2016, Dynamic electro-thermo-mechanical modelling of a U-shaped electro-thermal actuator, *J. Micromech. Microeng.* 26 (2016) 025010 (11pp).

[R5] Yi Zhu, Evgueni T Filipov, 2021, Rapid multi-physics simulation for electro-thermal origami systems, *International Journal of Mechanical Sciences*, 202, 106537

* How is the convection coefficient $h_{(p/f)}$ determined and calculated in the manuscript? It seems that it is not discussed anywhere in the manuscript as well as in the supplementary. Also, in the Equation (1) of the supplementary material, is the term Q_3 conduction based, or convection based? It looks like the provided equations are not checked thoroughly yet in the manuscript.

* The work claims major development on the simulation - "To achieve precise control of the electrothermal origami, we develop a multi-physics simulation model about the deploying process

by finite element analysis with the ABAQUS software.” Is this a new customizable element implemented in the ABAQUS? Or how is the algorithm and model implemented? In that case, please provide the associated simulation file, implementation package, and sufficient description in the SI so that the paper adheres to the guideline of Nature Communications regarding using codes and algorithms. On the other hand, if all simulations are easily implemented using internal elements in ABAQUS (a commercial software), perhaps the development is not significant enough to be seen as one major contribution. In this case, the authors may need to further develop new algorithms to “inverse design” for the precise control so that there is sufficient development.

* It seems that the simulation relies on curve fitting experimental results to function properly (Supplementary Fig 3). If the developed simulations require experiments in the first place to calibrate, how can we achieve “precise” control of the folding motion using the simulation? Why not directly control based on the measured experimental curves? Please discuss it in the paper.

Application Scenarios:

* Robot Gripper: Since the origami design is based on shape memory effects, is it true that the system cannot achieve repeatable motions? For example, for the gripper shown on Fig 4, using shape memory effects probably means that the gripper cannot release the object and redo the gripping motion without reprogramming. Please discuss the limitations of this system.

* Robot Gripper: The control of the electro-thermal origami is based on removing the alligator clip and manually reconnecting it with other wires. Have the authors considered how to resolve this problem in real applications? It looks like the exposed CCF wire outside the hinge region also generates heat. If these CCF wires are used as connection wire within an origami robot body, heating one crease may cause folding in another crease because the connection CCF wires are close to the other crease. For the paper to demonstrate convincing application as robotic grippers, these abovementioned limitations need to be fully addressed.

* Architected Structure: For the system developed in Fig. 6, the narration may need a second thought. For a system to be called metamaterial, it should be possible to “homogenize” the system as a “material” with certain material properties. This usually requires the system to contain many unit cells. Otherwise, it is better to refer to the system as architected structures. The authors have demonstrated a case with just nine units. Can the system be made to contain more units? Does increasing the number of units change the responses of the homogenized material property? How to add more units in the z-direction?

* Wing structure: Can the shape of the wing be continuously changed back and forth when in use? If not, why would engineers be interested in using this technology when there are many methods to create morphing wing structures that can continuously and repeatedly change their geometry in use [R6, R7]. Please discuss it in the manuscript.

[R6] Daochun Li, Shiwei Zhao, Andrea Da Ronch, et. al., 2018, A review of modelling and analysis of morphing wings, *Progress in Aerospace Sciences*, 100, 46-62.

[R7] Nguyen K. Pham and Edwin A. Peraza Hernandez, 2021, Modeling and Design Exploration of a Morphing Wing Enabled by a Twisting Tensegrity Mechanism, AIAA 2021-0099, DOI: 10.2514/6.2021-0099.

Reviewer #2 (Remarks to the Author):

This paper presents a new fabrication-design-actuation methodology for precisely controlled electrothermal origami with excellent mechanical performance and spatiotemporal controllability. By integrating continuous carbon fibers for Joule heating onto the hinges of the origami structure via FDM 4D printing, PCEO can change shape precisely through electrical actuation. CCFs can enhance the stiffness of pure SMP in the rubbery state and improve the uniformity of thermal

distribution at the hinged meanwhile. Reconfigurable robot gripper, mechanical-tunable Miura-origami unit, and airfoil-adjustable wing have been constructed to demonstrate the potential applications of the PCEO. This work is impressive and essential to obtain active origami devices and machines.

For this reason I support its publication by Nature Communications with the following comments to consider for revision.

Following are some major concerns.

1. How do the arrangement and geometry characteristics of CCFs effect the mechanical performance of the PCEO?
2. Authors should check the definition of the shape recovery ratio R_r in Fig 2e, and how does the hinge recovery force effect the deployment of PCEO? CCF-SMP at 0.30A presents a faster shape recovery process than at 0.26A, but a lower recovery force, how about other currents?
3. Could the shape-reconfigurable PCEO strip structure in Fig 4 transform among configuration 1-6? And the gripper functional configuration by sequence control seems can only be constructed once?
4. The variable thickness wing is a tessellation of five Miura-origami units, the author should demonstrate the unique advantages and advancements of the proposed wing comparing with the Miura-origami unit in Fig 5.

Reviewer #3 (Remarks to the Author):

I have attached a PDF version of my comments as well.

The paper presents the creation of origami structures through the use of muliti-matierial additive manufacturing of a shape memory polymer, in this case polylatic acid, and continuous carbon fiber. The carbon fiber material not only acts as a reinforcer, but also as a heating element that can be used to locally heat the shape memory polymer initiating the shape memory process. Advantages of this manufacturing methodology include the ability to stop the shape recovery process by stopping the applied current (thereby enabling the ability to program a multitude of temporary shapes), the ability to move different areas of a larger structure by differing the amount and duration of applied current discretely, and the ability to use the localized deformation capability to locally change the geometry effectively locally controlling the mechanical properties in a metamaterial structure.

Overall, the article is interesting. There is a large amount of supplementary data. I have gone through the supplied documents and videos which are also quite interesting. There are some issues with this article that should be remedied before it can be published There are some grammatical issues and aspects that could use elaboration, particularly in the Materials and Methods section. I have detailed these issues in la line by line manner below:

Line 17. First sentence of abstract should be rewritten.

Introduction. There are several other researchers who have made additively manufactured shape memory components with continuous carbon fibers as you have done. Please include references to other works and state how your work differs from theirs.

Line 103-line 105. It would be useful to state Joule's Law $Q=I^2Rt$ to give an idea of how much heat is being generated over the 30 second interval. It may be better to move Equation (1) on line 164 to earlier in the manuscript.

Line 149 it please correct $\text{Tan}\delta$ to $\tan \delta$. It may also be useful to indicate that $\tan \delta$ means loss tangent.

Line 183 is somewhat unclear. How is the permanent shape programmed? Generally with SMPs the programmed shape is the permanent shape.

The sentence says the permanent shape is programmed to the temporary shape. It should be rewritten similar to: the specimen is heated by joule heating and then deformed from the

permanent shape $\theta_{\max}=180$ to a temporary shape $\theta=0$ and then allowed to cool by powering off the system.

Is the specimen manually deformed to a temporary shape?

Line 186. Believe the equation for shape fixation should be on its own line and denoted as Equation 2 also the shape recovery equation should be Equation 3. Be sure to correct the numberings of your subsequent equations in the manuscript.

Line 191 I do not think that the equation used to calculate recovery ratio is correct. The first issue I see is that it looks like you are multiplying θ by time. This would effectively give a unit of time in the final value. I believe you are trying to say that the time dependent angle is the angle measured at a discrete time during the recovery process. Am I correct in this assumption? If this is the case, please denote the angle as θ_t with the t as a subscript. Also please explain what the time dependent angle is more clearly in the text. Figure 2 e indicates it as the measured angle after recovery. If it is the measured angle after recovery, please state this in the text or denote the angle at recovery at the final time or t_f or some other designation that is clear to the reader. Also, the other problem with the equation is that if the angle θ is equal to 1 your recovery ratio will be zero and if the value is greater than 1 your recovery ratio will be negative. Please check this equation or provide more explanation to this calculation in the text. The examples provided are complete. The documentation of this work is well done. The reconfiguration capability through localized reprogramming is interesting.

Materials and methods section should be written in the past tense.

Line 482 should be changed to: The SMP material used was polylactic acid. Similar grammatical issues are found throughout this section. Please correct.

Line 483 It is unclear why and how a support material was used during the printing of your specimens. The printer is dual extruder where one extruder printed PLA and the other the continuous carbon fiber. How was the material removed?

Line 490 "built" should be changed to "build"

Line 491. Should be changed to 297 mm x 201 mm x 147 mm.

Line 500 should be changed to : was used to record. There are several tense errors in this section as I have mentioned before.

Line 511 Again I think it is correct to write out as 250 mm x 15 mm by 1 mm. Please confirm journal style.

Line 524 sometimes you refer to a figure in text as Figure x, here you call it Fig. 1f. Please check document for consistency.

Lines 552 to Lines 555. Similar to previous comments, it is unclear if the specimen is manually moved to the temporary shape. How is the temporary shape held for five minutes? Is there a measured load applied to the specimen to hold it in this temporary shape. Please elaborate.

REVIEWER COMMENTS

Response: the authors would like to thank the editors and reviewers for their valuable comments which help us improve our paper significantly. We have carefully responded the reviewers' comments and suggestions point-by-point, and have revised our paper accordingly. Our responses to each of these comments are given as follows.

Comments from Reviewer #1

Comment 1.1: This paper introduces a method to build electro-thermal origami with continuous carbon fiber (CCF) and shape memory polymer (SMP). The paper shows fabrication method, simulation for precise control, and application scenarios of the proposed technology. The major claimed contributions include: (1) use of CCF to improve mechanical characteristics of SMP, (2) develop simulation for precise control, (3) demonstrate application potentials. Although the developments are interesting, the three aspects are not yet well-developed and convincing enough for a top-tier journal like Nature Communications. Please consider addressing the following comments before the publication can be accepted:

Response: we thank the reviewer for taking precious time to review our paper, offering constructive comments and suggestions, and providing a comprehensive summary of our work. In light of the referee's comments, we have revised the manuscript accordingly. We hope the amended version will convince the reviewer of its suitability for publication in *Nature Communications*.

In addition, we further clarify here the impacts and contributions of our work:

(i) We for the first time report a 4D printing strategy of continuous fiber-reinforced shape memory polymer (SMP) to fabricate **complex active origami in a single step**. We enhance the modulus of SMP by 1000 times in the rubbery state, from **2.0 MPa to 2.3 GPa**. The excellent Joule heating characteristics and high thermal conductivity of continuous carbon fibers endow **a uniform thermal distribution with rapid heating** at the hinge, which guarantees the larger shape recovery ratio and force.

(ii) We develop a **multi-physics model** that considers **electrical, thermal, and mechanical** effects for the deployment control process of PCEO. This model allows us to **precisely deploy the reconfigurable electrothermal origami in both time and space**.

(iii) The **precise control capability** of electrothermal origami allows us to create structures and architected materials with **multifunctionalities**. We design and fabricate reconfigurable robots, customizable architected materials, and programmable wings.

These unique demonstrations have not been founded in other studies on 4D printing of continuous fiber-reinforced shape memory composites.

Fabrication of CCF within SMP:

Comment 1.2: Embedding continuous carbon fiber (or other continuous fiber) in SMP (or other polymer) composite is not a brand-new idea. Instead, it was demonstrated in previous research [R1-R3]. For example, in the paper by Wang et. al. [R1], they have already demonstrated the idea of combining continuous fiber and 4D printing to build multi-functional systems. Applying a well-developed fabrication to build origami may not be itself an innovative enough idea for Nature Communications. Is there improvement in the techniques? Or is there other worth mentioning achievement? Please consider providing more discussion regarding the fabrication.

[R1] Qingrui Wang, Xiaoyong Tian, Lan Huang, et. al., 2018, Programmable morphing composites with embedded continuous fibers by 4D printing. *Materials and Design*, 155, 404–413.

[R2] Ryosuke Matsuzaki, Masahito Ueda, Masaki Namiki, et. al., 2016, Three-dimensional printing of continuous-fiber composites by in-nozzle impregnation, *Scientific Report*, 6:23058.

[R3] Chengjun Zeng, Liwu Liu, Wenfeng Bian, et. al., 2020, 4D printed electro-induced continuous carbon fiber reinforced shape memory polymer composites with excellent bending resistance, *Composites Part B*, 194, 108034.

Response 1: we appreciate the reviewer for this comment. In the revision, we have added one paragraph in Introduction to discuss our contribution in contrast to other related works, including those suggested by the reviewer.

Action: the added paragraph in Introduction reads:

“Recent advancements in 3D printed continuous fiber-reinforced composites^{31, 32} offer an attractive solution to the issues when using SMP-based 4D printing to realize active origami. These composites demonstrate superior mechanical properties than composites with other reinforcement forms³³⁻³⁶. Additionally, an improved global shape-shifting ability through Joule heating³⁷⁻⁴⁰ of continuous conductive fibers in SMP matrices has been demonstrated with simple structures^{37, 38} or classical lightweight designs, e.g., 2D cellular structures^{39, 40}. However, despite this promising solution, several challenges in the 4D printing of continuous fibers-reinforced composites must be addressed to successfully create active origami with high load-bearing capacity and precise control. Firstly, the design of rigid-foldable origami necessitates a more intricate fiber layout to engineer local stiffness. Moreover, the multi-physical and highly nonlinear deploying process of these composites remains poorly understood. There is also a need to establish related modeling and control methods, which are pivotal for

the precise deployment of active origami, ultimately enabling on-demand modulation of its properties for multi-scenario and multi-take applications.”

Response 2: to distinguish our work from references [R1-R3] (References 31, 32, and 38 in the revision respectively), we present Table R1 and Figure R1:

		Reference [R1]’s work	Reference [R2]’s work	Reference [R3]’s work	This work
Structural complexity		Thin curved surfaces	Rectangular specimens	Rectangular specimens	Complex origami
Roles of continuous carbon fiber	Reinforcement	√	√	√	√
	Heat conduction	-	-	√	√
	Localized Joule heating	Global heating	-	Global heating	Localized heating
Investigation of the heating process of fibers		Experiment	-	Experiment	Experiment and theory
Investigation on the impact of fibers on thermal distribution		-	-	Experiment	Experiment and simulation
Deployment process control	Unfolding angle	-	-	-	√
	Local	-	-	-	√
	Sequence	-	-	-	√
Number of states		Two states	-	Two states	Multiple states
Multi-functional systems	Programmable geometry	√	-	-	√
	Tunable mechanical properties	-	-	-	√
	Reconfigurable gripper	-	-	-	√

Table R1. Comparison of the 3D/4D printed continuous carbon fiber-reinforced composites presented in this work and references suggested by Reviewer 1.

- (i) This work proposes a 4D printing strategy of continuous carbon fiber-reinforced shape memory polymer to fabricate **complex active origami** in a single step (Figure R1c). In contrast, reference [R1] mainly focuses on the design and 4D printing of thin curved surfaces (Figure R1d and R1e). Meanwhile, references [R2] and [R3] are primarily focused on the 3D/4D printing of rectangular specimens (Figure R1f and R1g).
- (ii) In this work, the continuous carbon fibers (CCFs) within the active origami serve **three key roles**: CCFs in the stiff panels and hinges for **reinforcement**, CCFs in the hinges for **heat conduction**, and CCFs on the hinges for **localized Joule heating** (Figure R1a). In comparison, reference [R1] primarily focuses on the reinforcement and global Joule heating roles of CCFs, reference [R2] concentrates mainly on the reinforcement role of CCFs, and reference [R3] explores the roles of CCFs for reinforcement, global Joule heating and heat conduction. More importantly, this work **theoretically analyzes the highly nonlinear heating process of CCFs and simulates the impact of CCFs on thermal distribution**. In

contrast, references [R1] and [R3] mainly studied the relationship between power and temperature through experiments, reference [R2] dose not delve into the electrothermal properties of CCFs.

- (iii) This work develops a **multi-physics model** that considers **electrical, thermal, and mechanical** effects for the **deployment control process** of precisely controlled electrothermal origami (PCEO). We study the effect of Joule heating time on the shape-shifting process through **simulation** and prove that the **shape-shifting process can be precisely controlled by exploiting rapid cooling via power outage**. This model allows us to **precisely deploy the electrothermal origami** in both **time** and **space** with **unfolding angle, local, and sequence controllability**. In contrast, references [R1] and [R3] mainly studied the electro-induced shape-shifting behavior from one state to another state through experiments, without the study on the control of the shape-shifting process (Figure R1d-g). Reference [R2] has not studied the shape-shifting of continuous carbon fiber-reinforced composites.

- (iv) The precise control capability of electrothermal origami enables us to create **structures and architected materials with multifunctionalities**. We design and manufacture reconfigurable robots, customizable architected materials, and programmable wings. These unique demonstrations have not been founded in other studies on 4D printing of continuous fiber-reinforced shape memory composites.

Figure R1. Comparison of the 3D/4D printed continuous carbon fiber-reinforced composites presented in this work and references. **a-c** 4D printed precisely controlled electrothermal origami presented in this work. **d-e** 4D printed programmable surface composites mentioned in reference [R1]. **f** Improved mechanical property of 3D printed continuous carbon fiber-reinforced composites mentioned in reference [R2]. **g** 4D printed electro-induced continuous carbon fiber-reinforced composites mentioned in reference [R3].

Comment 1.3: Following the previous comment, it is necessary to present a more detailed literature review regarding how prior research uses 3D/4D printing together with continuous fibers. This will help readers to identify how this work is different from prior research.

Response: we thank the reviewers for the constructive suggestion. We have added one paragraph in Introduction to discuss related works about 3D/4D printing of continuous fiber-reinforced shape memory composites to help readers identify how this work is different from prior research.

Action: the added paragraph in Introduction reads:

“Recent advancements in 3D printed continuous fiber-reinforced composites^{31, 32} offer an attractive solution to the issues when using SMP-based 4D printing to realize active origami. These composites demonstrate superior mechanical properties than composites with other reinforcement forms³³⁻³⁶. Additionally, an improved global shape-shifting ability through Joule heating³⁷⁻⁴⁰ of continuous conductive fibers in SMP matrices has been demonstrated with simple structures^{37, 38} or classical lightweight designs, e.g., 2D cellular structures^{39, 40}. However, despite this promising solution, several challenges in the 4D printing of continuous fibers-reinforced composites must be addressed to successfully create active origami with high load-bearing capacity and precise control. Firstly, the design of rigid-foldable origami necessitates a more intricate fiber layout to engineer local stiffness. Moreover, the multi-physical and highly nonlinear deploying process of these composites remains poorly understood. There is also a need to establish related modeling and control methods, which are pivotal for the precise deployment of active origami, ultimately enabling on-demand modulation of its properties for multi-scenario and multi-task applications.”

References:

31. Matsuzaki R, et al. Three-dimensional printing of continuous-fiber composites by in-nozzle impregnation. *Sci Rep* **6**, 23058 (2016).
32. Wang Q, Tian X, Huang L, Li D, Malakhov AV, Polilov AN. Programmable morphing composites with embedded continuous fibers by 4D printing. *Mater Des* **155**, 404-413 (2018).
33. Zeng C, Liu L, Bian W, Leng J, Liu Y. Temperature-dependent mechanical response of 4D printed composite lattice structures reinforced by continuous fiber. *Compos Struct* **280**, 114952 (2022).
34. Dong K, Wang Y, Wang Z, Qiu W, Zheng P, Xiong Y. Reusability and energy absorption behavior of 4D printed continuous fiber-reinforced auxetic composite structures. *Composites, Part A* **169**, 107529 (2023).
35. Zeng C, Liu L, Bian W, Leng J, Liu Y. Compression behavior and energy absorption of 3D printed continuous fiber reinforced composite honeycomb structures

with shape memory effects. *Addit Manuf* **38**, 101842 (2021).

36. Dong K, Ke H, Panahi-Sarmad M, Yang T, Huang X, Xiao X. Mechanical properties and shape memory effect of 4D printed cellular structure composite with a novel continuous fiber-reinforced printing path. *Mater Des* **198**, 109303 (2021).

37. Chen H, et al. Electrothermal shape memory behavior and recovery force of four-dimensional printed continuous carbon fiber/polylactic acid composite. *Smart Mater Struct* **30**, 025040 (2021).

38. Zeng C, Liu L, Bian W, Liu Y, Leng J. 4D printed electro-induced continuous carbon fiber reinforced shape memory polymer composites with excellent bending resistance. *Composites, Part B* **194**, 108034 (2020).

39. Ye W, Dou H, Cheng Y, Zhang D. Self-sensing properties of 3D printed continuous carbon fiber-reinforced PLA/TPU honeycomb structures during cyclic compression. *Mater Lett* **317**, 132077 (2022).

40. Dong K, Panahi-Sarmad M, Cui Z, Huang X, Xiao X. Electro-induced shape memory effect of 4D printed auxetic composite using PLA/TPU/CNT filament embedded synergistically with continuous carbon fiber: A theoretical & experimental analysis. *Composites, Part B* **220**, 108994 (2021).

Simulation of 4D printing of origami for precise control:

Comment 1.4: It seems that the author is assuming no conduction between the folding crease and the surrounding atmosphere as well as the panels (as forms of heat loss). However, for small-scale electro-thermal systems, conduction is usually the major form of heat loss [R4, R5]. For the scale of the reported electro-thermal origami system, it is likely that conduction is still contributing to a considerable part of the heat loss. Can the authors justify their assumption by including conduction in the model and show that conduction is indeed not the major contributor?

[R4] Hussein Hussein, Aref Tahhan, Patrice Le Moal, et. al., 2016, Dynamic electro-thermo-mechanical modelling of a U-shaped electro-thermal actuator, *J. Micromech. Microeng.* 26 (2016) 025010 (11pp).

[R5] Yi Zhu, Evgueni T Filipov, 2021, Rapid multi-physics simulation for electro-thermal origami systems, *International Journal of Mechanical Sciences*, 202, 106537

Response 1: we appreciate the reviewer for this comment. We **do consider the heat loss between the hinge and the surrounding atmosphere as well as the panels**, which is the key to our precise control of electrothermal origami (Figure R2a, Figure 3a in Manuscript). However, our model considers **heat convection as the predominant form of heat transfer** between the hinge and the surrounding atmosphere. To study the

impact of convection between the hinge and the surrounding environment on the heat transfer of our electrothermal origami system, we **compared the heating curves** of the hinge and its **heat distribution with and without considering convection** through simulations. Figure R2b presents the simulation results, which closely match the experimental results when convection is considered. More specifically, without considering convection, the hinge’s temperature continues to rise due to the lack of heat loss. Eventually, the temperature of the hinge is up to 700 °C under 0.26 A Joule heating for 40 seconds, a situation clearly divergent from reality. Therefore, **heat convection is a very important factor** in simulating the heat transfer of the electrothermal origami system. In the revision, we have added additional discussions to underscore the importance of convection and added Figures R2b and R2c as Figure 12 in Supplementary Information.

Action: the added text in Line 278 in the revision and figure (Supplementary Figure 12) are as follows:

“It should be noted that considering heat loss through convection is crucial for the heat transfer analysis of the electrothermal origami (Supplementary Fig.12).”

Figure R2. Comparison of the hinge thermal distribution with and without considering convection. a (Figure 3a in Manuscript) The heat transfer model for the proposed PCEO. Comparison of the hinge heating (b) and thermal distribution (c) with and without considering convection under 0.26 A Joule heating (Figure 12 in Supplementary Information).

Response 2: we appreciate the two references provided by the reviewer. After carefully reading these references, we noted that both primarily focused on micro-electromechanical systems (MEMS), where **free convection is negligible compared to conduction** (Hussein et al., 2016, Zhu et al., 2021). However, at the **macroscale**,

free convection would be the dominant factor (Ozsun et al., 2009). In this study, our electrothermal origami is on the centimeter scale, classifying it as a macroscale system.

References:

Hussein H, Tahhan A, Le Moal P, Bourbon G, Haddab Y, Lutz P. Dynamic electro-thermo-mechanical modelling of a U-shaped electro-thermal actuator. *Journal of Micromechanics and Microengineering* **26**, 025010 (2016).

Zhu Y, Filipov ET. Rapid multi-physics simulation for electro-thermal origami systems. *International Journal of Mechanical Sciences* **202-203**, 106537 (2021).

Ozsun O, Alaca BE, Yalcinkaya AD, Yilmaz M, Zervas M, Leblebici Y. On heat transfer at microscale with implications for microactuator design. *Journal of Micromechanics and Microengineering* **19**, 045020 (2009).

Response 3: in addition, we **do consider the heat conduction between the hinge and panels**. We mentioned in the submitted manuscript that “Figure 3a depicts the electro-thermal processes considered in the simulation, including Joule heating of CCFs, **heat conduction in hinges and stiff panels**, and heat convection between air and hinges and stiff panels.”. As shown in Figures R3a and R3b (Figures 3c and 3f in Manuscript), the hinge conducts heat to the panel (as a form of heat loss), showing that the part of the panel near the hinge is heated up.

Figure R3. Demonstration of considering heat conduction between hinges and panels. **a** (Figure 3c in Manuscript) Simulated temperature distribution and the unfolding angle at power off and after convection cooling for 10 seconds of the CCF-SMP hinge under different heating times. **b** (Figure 3f in Manuscript) Programming of the temporary shape and actively controlled recovery of the PS of the airplane-shaped

PCEO structure (LS1, LS2, and LS3 achieved by connecting both end electrodes of hinges 1, 3, and 1-2-3 at a time) realized by FEA.

Comment 1.5: How is the convection coefficient h (p/f) determined and calculated in the manuscript? It seems that it is not discussed anywhere in the manuscript as well as in the supplementary. Also, in the Equation (1) of the supplementary material, is the term Q_3 conduction based, or convection based? It looks like the provided equations are not checked thoroughly yet in the manuscript.

Response 1: we thank the reviewer for the constructive comment. To avoid any confusion, we have modified Equation (6) to $\dot{\Phi}_{\text{CONV}} = \frac{hA}{V}(T_s - T_a)$, where h is the convection coefficient between air and origami surface. For convection between air and fibers, h is denoted as h_F . For convection between air and polymers, h is denoted as h_P . First, we determined that the convection coefficient of fibers h_F is $0.065 \text{ mW}/(\text{mm}^2 \cdot \text{K})$ by simulating the heating curves of CCFs under different convection coefficients and comparing them with experimental and numerical results (Figure R4a). Then, we determined that the convection coefficient of polymers h_P is $0.020 \text{ mW}/(\text{mm}^2 \cdot \text{K})$ by simulating the heating curves of hinges under different convection coefficients and comparing them with the experimental results (Figure R4b). Moreover, we checked the correctness of the identified convection coefficients by examining the difference between simulation and experimental results of heating curves of hinges under other current stimuli (Figure R4c). In the revision, we have added a few discussions on the determination of the convection coefficient and added Figure 11 in Supplementary Information.

Action 1: the added text in Line 275 in the revision and figure (Supplementary Figure 11) are as follows:

“The heat loss through convection per unit time and unit volume is expressed as $\dot{\Phi}_{\text{CONV}} = \frac{hA}{V}(T_s - T_a)$, where h is the convection coefficient (More details about the determination of h are presented in Supplementary Fig. 11),”

Figure R4 (Figure 11 in Supplementary Information). The determination of the convection coefficient. It should be noted that the convection coefficient between air

and different materials varies. For convection between air and fibers, h is denoted as h_F . For convection between air and polymers, h is denoted as h_P . **a** The determination of the convection coefficient of fibers by simulating the heating of CCFs with an applied current of 0.26 A. **b** The determination of the convection coefficient of polymers by simulating the heating of hinges with an applied current of 0.26 A. **c** The validation of their effectiveness. It should be noted that the heat transfer between the structure and the surrounding environment is simplified as a heat convection problem. The convection coefficient of fibers is determined to be $0.065 \text{ mW}/(\text{mm}^2 \cdot \text{K})$ and that of polymers is $0.020 \text{ mW}/(\text{mm}^2 \cdot \text{K})$.

Response 2: regarding to the comment related to Q_3, the heat transfer between continuous carbon fibers (CCFs) and air during Joule heating is mainly through **heat convection**. Therefore, in Equation (1) of the Supplementary Information, the term Q_3 is **convection-based**. In the revision, we have added corresponding explanations in Supplementary Information.

Action 2: the modified text in the last sentence of the second paragraph which is highlighted in yellow color in the Supplementary Information reads:

*“According to the heat balance principle (i.e., thermal equilibrium), the heat Q_1 generated by Joule heating of CCFs is equal to the sum of the heat consumption Q_2 and **convection-based** heat dissipation Q_3 .”*

Comment 1.6: The work claims major development on the simulation - “To achieve precise control of the electrothermal origami, we develop a multi-physics simulation model about the deploying process by finite element analysis with the ABAQUS software.” Is this a new customizable element implemented in the ABAQUS? Or how is the algorithm and model implemented? In that case, please provide the associated simulation file, implementation package, and sufficient description in the SI so that the paper adheres to the guideline of Nature Communications regarding using codes and algorithms. On the other hand, if all simulations are easily implemented using internal elements in ABAQUS (a commercial software), perhaps the development is not significant enough to be seen as one major contribution. In this case, the authors may need to further develop new algorithms to “inverse design” for the precise control so that there is sufficient development.

Response: we thank the reviewer for the comment. In the submitted manuscript, we explained that the implemented element is a built-in type in ABAQUS, as stated in the Methods section of the Manuscript: “The structure was modeled using an eight-node linear heat transfer brick element (ABAQUS element type DC3D8) for CCF-SMP and a ten-node bilinear quadratic heat transfer tetrahedron element (ABAQUS element type DC3D10) for SMP.”

The establishment of the multi-physics model and precise simulation of the shape-shifting behavior of the electrothermal origami are **highly intricate** and **technically challenging** processes of **multi-physics field coupling**. Firstly, we need to numerically analyze the temperature rise of CCFs under Joule heating, where the fiber resistance exhibits **highly nonlinear** characteristics, i.e., changes with temperature (Figure R5a). Secondly, we need to establish a **multi-branch thermoviscoelastic model** that can capture the shape memory behavior of the shape memory composites, where the **relaxation time of each branch varies with temperature changes** (Figure R5b). Thirdly, we need to establish a multi-physics model, import the body heat flux of CCFs and material model into it, and determine their boundary conditions. Figure R5c depicts the electro-thermal processes considered in the model, including **Joule heating** of CCFs, **heat conduction** in hinges and stiff panels, and **heat convection** between air and hinges and stiff panels. Then, we need to simulate the shape-shifting behavior of the electrothermal origami with different inputs, the simulating process includes **heating, loading, cooling, unloading, reheating (recovering), and recooling (locking)**. Fourthly, we obtained the **locked unfolding angle of the electrothermal origami with a heating time from 0 s to 40 s** through simulation with a time interval of 2.5 s, the **locking angle range can be from 0° to nearly 180°** (Figure R5d). Through the simulation results, we can obtain the **guiding model for the heating time corresponding to the locking angle** (Figure R5d). Lastly, we have achieved **precise control** of the airplane-shaped origami with **complex deploying routines** and the strip origami with **reconfigurability** through our guiding model, where **all the heating times** corresponding to the target angles of each hinge are **derived from the guiding model** (Figure R5e). Therefore, we believe that our simulation model can provide sufficient guidance for precisely controlled electrothermal origami.

Figure R5. Establishment and guidance of the multi-physics model for precisely controlled electrothermal origami. **a** Numerical calculation of the highly nonlinear heating process of CCFs. **b** Establishment of the multi-branch thermoviscoelastic model for shape memory composites and the determination of the material parameters. **c** Import of the body heat flux of numerical calculation of CCFs and the material model, establishment of the multi-physics model, and simulation of the shape-shifting behaviors of electrothermal origami with different inputs. **d** Simulation results of the locked shape of the electrothermal origami with a heating time from 0 s to 40 s with a time interval of 2.5 s, and the establishment of the guiding model for the heating time corresponding to the locking angle. **e** Precise control of the airplane-shaped origami

with complex deploying routines and the strip origami with reconfigurability guided by the guiding model.

Comment 1.7: It seems that the simulation relies on curve fitting experimental results to function properly (Supplementary Fig 3). If the developed simulations require experiments in the first place to calibrate, how can we achieve “precise” control of the folding motion using the simulation? Why not directly control based on the measured experimental curves? Please discuss it in the paper.

Response: we thank the reviewer for this comment. To provide a clearer understanding of how the multi-physics simulation works, we present its flowchart as shown in Figure R6. The process includes nine steps: i) measuring the necessary parameters of CCFs for numerical calculation of its heating curve; ii) calculating the body heat flux of CCFs based on their heating curves under different currents; iii) importing the body heat flux of CCFs into the finite element model established for the heat transfer simulation of the electrothermal origami, and determining the boundary conditions (convection coefficients) of the model; iv) validating the effectiveness of the heat transfer model by comparing the hinge heating results of simulation and experiment under different currents; v) importing the multi-branch thermoviscoelastic model for shape memory composites into the simulation model, and creating the shape memory composites programming process; vi) validating the multi-physics model by comparing the shape recovery results of simulation and experiment; vii) simulating the shape-shifting behavior of the electrothermal origami under different heating times; viii) generating the guiding model for the heating time corresponding to the locking angle through simulation, and validating its effectiveness by conducting experiments; ix) guiding the precise control of the electrothermal origami by validated model.

Regarding to the comment “It seems that the simulation relies on curve fitting experimental results to function properly (Supplementary Fig 3).”, **physical quantities** such as the resistivity, temperature coefficient of resistance, and overall heat transfer coefficient of CCFs (the first step in Figure R6, Supplementary Fig 3 in the submitted file) are **essential model inputs to be experimentally characterized**. The accuracy of these inputs ensures the model’s ability to **precisely predict the heating curves for untested conditions** through simulations. Regarding to the comment “If the developed simulations require experiments in the first place to calibrate, how can we achieve “precise” control of the folding motion using the simulation? Why not directly control based on the measured experimental curves?”, firstly, we **validated the effectiveness of the heat transfer model** by comparing the hinge heating results of simulation and experiment under different currents; secondly, we **validated the multi-physics model** by comparing the shape recovery results of simulation and experiment; finally, we **validated the guiding model** for the heating time corresponding to the unfolding angle by conducting experiments. The final step in the flowchart of multi-physics simulation presents that our model can effectively guide the precise control of electrothermal origami. Therefore, we can achieve precise control of the folding motion using the

simulation.

Figure R6. Flowchart of multi-physics simulation for precisely controlled electrothermal origami.

Application Scenarios:

Comment 1.8: Robot Gripper: Since the origami design is based on shape memory effects, is it true that the system cannot achieve repeatable motions? For example, for the gripper shown on Fig 4, using shape memory effects probably means that the gripper cannot release the object and redo the gripping motion without reprogramming. Please discuss the limitations of this system.

Response: we thank the reviewer for this valuable question. The shape memory polymer used in this work is a one-way memory material, which means that the reuse of the gripper functional configuration requires reprogramming. In the revision, we have incorporated additional text in the Reconfigurability of the PCEO and Discussion sections of the Manuscript to address the limitations of this system.

Action: the added text in Line 361 and Line 510 in the revision reads:

“It should be noted that all configurations originate from one single PCEO strip, and its reconfigurable deployment requires reprogramming.”

“However, considerable efforts are needed to translate these design concepts, e.g., the reconfigurable robot gripper and the variable thickness wing, into real-world applications. For instance, reversible deployment control can be realized by the hybrid-actuation solution including cable driven, etc., as an alternative to the bulky actuation units required by the non-responsive material system. Moreover, precise control under complicated environments, e.g., dynamic loads and extreme temperatures, deserves further study to align with specific functional requirements.”

Comment 1.9: Robot Gripper: The control of the electro-thermal origami is based on removing the alligator clip and manually reconnecting it with other wires. Have the authors considered how to resolve this problem in real applications? It looks like the exposed CCF wire outside the hinge region also generates heat. If these CCF wires are used as connection wire within an origami robot body, heating one crease may cause folding in another crease because the connection CCF wires are close to the other crease. For the paper to demonstrate convincing application as robotic grippers, these abovementioned limitations need to be fully addressed.

Response: we thank the reviewer for the very constructive comment. To achieve automatic control of the reconfigurable PCEO, we have built a control system as shown in Figure R7a. The control system consists of two parts: the power supply and electrical control systems. The voltage was provided by a regulated 24-volt supply. The switches of the hinges are controlled by the signal from the microcontroller via relays. Moreover, we have insulated the connection between fibers and wires through heat shrink tubing. In fact, heating a hinge will not affect adjacent hinges because the distance between them is relatively far (> 2 cm). Figure R7b presents the deployment process of PCEO automatically realized by the control system, which demonstrates the feasibility of using the self-built control system to precisely control the unfolding process of PCEO. In the revision, we have added text to discuss the future work.

Action: the added text in Line 510 in the revision reads:

“However, considerable efforts are needed to translate these design concepts, e.g., the reconfigurable robot gripper and the variable thickness wing, into real-world applications. For instance, reversible deployment control can be realized by the hybrid-actuation solution including cable driven, etc., as an alternative to the bulky actuation units required by the non-responsive material system. Moreover, precise control under complicated environments, e.g., dynamic loads and extreme temperatures, deserves further study to align with specific functional requirements.”

Figure R7. Control of the reconfigurable PCEO. **a** Control system of the reconfigurable PCEO. The control system consists of two parts: the power supply and control systems. The voltage was provided by a regulated a regulated 24-volt supply (GPD-4303S, GWinstek, China). The switches of the hinges are controlled by the signal from the microcontroller (Arduino UNO R3, DFROBOT, China) via relays (TOUGLESY, CHNT, China). **b** Deployment process of PCEO automatically realized by the self-built control system, Scale bar, 20 mm.

Comment 1.10: Architected Structure: For the system developed in Fig. 6, the narration may need a second thought. For a system to be called metamaterial, it should be possible to “homogenize” the system as a “material” with certain material properties. This usually requires the system to contain many unit cells. Otherwise, it is better to refer to the system as architected structures. The authors have demonstrated a case with just nine units. Can the system be made to contain more units? Does increasing the number of units change the responses of the homogenized material property? How to add more units in the z-direction?

Response: we appreciate the reviewer for the very valuable comment. In the revision,

we have revised “Metamaterial” as “Architected materials”. As shown in Figure R8, the architected material can be **expanded in the X and Y directions** by plugging digital units into the expanded bottom plate with a periodically arranged stopper array. The combinatory digital architected material can effectively **decouple the deformation constraints between interconnected building blocks**, which greatly increases the deformability and programmability of materials. The mechanical properties of the combinatory digital architected material can be obtained by **linearly superposing the compressive force-displacement curves of these decoupled units**. As shown in the example in Figure R9, when Combination 1 is expanded from 9 units (two Ori_{1S}, two Ori_{2S}, and five Ori_{3S}) to 18 units (four Ori_{1S}, four Ori_{2S}, and ten Ori_{3S}), the force in the force-displacement curve doubles (Figure R9a) while the stress-strain curve remains unchanged (Figure R9b). Moreover, Figure R8 depicted that the architected material can also be **expanded in the Z direction** by stacking up in a layer-wise manner. It should be noted that the distribution of units with different configurations in each layer should be as uniform as possible to keep the center of gravity of each layer close to the centroid. If there is only one unit with the highest height configuration on a certain layer, it needs to be placed at the center of the layer to maintain the architected material balance through its two top ridges. The texts have been revised and added to the Manuscript and Supplementary Figure 22 (Figure R8) has been added to Supplementary Information.

Action: the revised text in Line 410 and Line 431 in the revision and the added figure (Supplementary Figure 22) are as follows:

*“The programmability of architected materials refers to the ability to achieve the on-demand modulation of mechanical property **by producing nonuniformly distributed regions of deformation**⁴³⁻⁴⁶.”*

*“We fabricate a 3×3 array architected material prototype based on Ori₁, Ori₂, and Ori₃ units to validate the feasibility of customizing the combinatory **digital architected materials, which can be further expanded in X, Y, and Z directions** (Supplementary Fig. 22).”*

Figure R8 (Figure 22 in Supplementary Information). The expansion of the combinatory digital architected material in X, Y, and Z directions. It should be noted that the distribution of units with different configurations in each layer should be as uniform as possible to keep the center of gravity of each layer close to the centroid. If there is only one unit with the highest height configuration on a certain layer, it needs to be placed at the center of the layer to maintain the architected material balance through its two top ridges.

Figure R9. The force-displacement curves (a) and stress-strain curves (b) of the architected material with 9 and 18 units in Combination 1.

References:

43. Overvelde JTB, Weaver JC, Hoberman C, Bertoldi K. Rational design of reconfigurable prismatic architected materials. *Nature* **541**, 347-352 (2017).
44. Shaw LA, Chizari S, Dotson M, Song Y, Hopkins JB. Compliant rolling-contact

architected materials for shape reconfigurability. *Nat Commun* **9**, 4594 (2018).

45. Chen T, Pauly M, Reis PM. A reprogrammable mechanical metamaterial with stable memory. *Nature* **589**, 386-390 (2021).

46. Ye H, et al. Multimaterial 3D printed self-locking thick-panel origami metamaterials. *Nat Commun* **14**, 1607 (2023).

Comment 1.11: Wing structure: Can the shape of the wing be continuously changed back and forth when in use? If not, why would engineers be interested in using this technology when there are many methods to create morphing wing structures that can continuously and repeatedly change their geometry in use [R6, R7]. Please discuss it in the manuscript.

[R6] Daochun Li, Shiwei Zhao, Andrea Da Ronch, et. al., 2018, A review of modelling and analysis of morphing wings, *Progress in Aerospace Sciences*, 100, 46-62.

[R7] Nguyen K. Pham and Edwin A. Peraza Hernandez, 2021, Modeling and Design Exploration of a Morphing Wing Enabled by a Twisting Tensegrity Mechanism, *AIAA 2021-0099*, DOI: 10.2514/6.2021-0099.

Response: we thank the reviewer for the constructive comment. The shape memory material used in this work is a one-way memory material, reversible shape deformation of the wing requires reprogramming. Moreover, wing programming can also be achieved by adding a small motor to drive the cable, thereby morphing back and forth continuously and repeatedly. However, future aircraft designs need to have improved fuel efficiency to meet the demand of the engineering application (Pham et al., 2021). **Bulky actuation systems and their tedious manufacturing processes** (Li et al., 2018), such as motors and pneumatic pumps, hinder the development of morphing wings. The wing in this work shows our capability to **rapidly construct complex origami-based morphing wings with precise deployment control in one step** (Figure R10). Additionally, we develop a model for the **inverse design of the airfoils**. Although using responsive materials to construct morphing wings presents a promising trajectory, there is still a long way to go before its practical engineering application. In the revision, we have added the text and Supplementary Figure 23 (Figure R10) to discuss the complexity and limitations of the variable thickness wing.

Action: the added text in Line 462, Line 471, and Line 510 in the revision and figure (Supplementary Figure 23) are as follows:

*“As illustrated in Fig. 7c, the mode of the variable thickness wing, with a fixed geometric design, can be altered by varying the heating time, **which also demonstrates that the proposed electrothermal control method can be extended to complex-shaped origami.**”*

“Nevertheless, a more challenging task to be explored is the inverse design of the entire cross-sectional profile of the airfoil, rather than focusing solely on its relative peak point position. This requires a more advanced computational-based method. Also, the precise deployment control of the skin is often indispensable in practical applications to conform to the airfoil of the wing, enabling both large deformation and shape-locking.”

“However, considerable efforts are needed to translate these design concepts, e.g., the reconfigurable robot gripper and the variable thickness wing, into real-world applications. For instance, reversible deployment control can be realized by the hybrid-actuation solution including cable driven, etc., as an alternative to the bulky actuation units required by the non-responsive material system. Moreover, precise control under complicated environments, e.g., dynamic loads and extreme temperatures, deserves further study to align with specific functional requirements.”

Figure R10 (Figure 23 in Supplementary Information). Construction and dimensions of the variable thickness wing based on PCEM. a Oblique view of the fiber path of the variable thickness wing. The first layer and the last layer are respectively the valley and mountain hinge layer, and the second and fifth layers are CCFs layers, heating for the valley and mountain hinge layer, respectively. **b** Vertical view of fiber path and dimension of the variable thickness wing.

References:

Pham NK, Peraza Hernandez EA. Modeling and Design Exploration of a Morphing Wing Enabled by a Twisting Tensegrity Mechanism. *In: AIAA Scitech 2021 Forum*. American Institute of Aeronautics and Astronautics (2021).

Li D, et al. A review of modelling and analysis of morphing wings. *Progress in Aerospace Sciences* **100**, 46-62 (2018).

Comments from Reviewer #2

Comment 2.1: This paper presents a new fabrication-design-actuation methodology for precisely controlled electrothermal origami with excellent mechanical performance and spatiotemporal controllability. By integrating continuous carbon fibers for Joule heating onto the hinges of the origami structure via FDM 4D printing, PCEO can change shape precisely through electrical actuation. CCFs can enhance the stiffness of pure SMP in the rubbery state and improve the uniformity of thermal distribution at the hinged meanwhile. Reconfigurable robot gripper, mechanical-tunable Miura-origami unit, and airfoil-adjustable wing have been constructed to demonstrate the potential applications of the PCEO. This work is impressive and essential to obtain active origami devices and machines.

For this reason I support its publication by Nature Communications with the following comments to consider for revision.

Response: we thank the reviewer for the constructive comments on our work. In light of the referee's comments, we have revised the manuscript accordingly.

Comment 2.2: How do the arrangement and geometry characteristics of CCFs effect the mechanical performance of the PCEO?

Response: we appreciate the reviewer's valuable comments. The anisotropic mechanical properties of continuous fibers make fiber arrangement a critical factor to consider in the design of composites. We performed tensile tests on three composite samples with different fiber angles (0°, 45°, 90°), and the tensile results are shown in Figure R11a. The specimen with a fiber angle of 0° (the fiber angle we used in this work) shows the higher tensile strength compared to the other two specimens. In addition, we also examined the effect of the fiber spacing (1.8, 2.9, 7.2 mm) on the mechanical properties of the specimen, and the results are shown in Figure R11b. It is found that the mechanical properties increase with the decrease in fiber spacing. Thus, this work utilizes the CCF-SMP specimen with 1.8 mm fiber spacing, which is the down limit of the selected manufacturing process. The effects of fiber arrangement parameters on CCF-SMP structures have been added to the Manuscript and Supplementary Information.

Action: the added text in Line 123 in the revision and figure (Supplementary Figure 2) are as follows:

“Supplementary Fig. 2 also presents the effect of fiber arrangement parameters (such as fiber angle and spacing) on the mechanical property of CCF-SMP.”

Figure R11 (Figure 2 in Supplementary Information). Tensile results of specimens with different fiber arrangement parameters. a Stress-strain curves of CCF-SMP specimen with different fiber angles. **b** Stress-strain curves of CCF-SMP specimen with different fiber spacing.

Comment 2.3: Authors should check the definition of the shape recovery ratio R_r in Fig 2e, and how does the hinge recovery force effect the deployment of PCEO? CCF-SMP at 0.30A presents a faster shape recovery process than at 0.26A, but a lower recovery force, how about other currents?

Response 1: we appreciate the reviewer for this important question. We have checked and revised the definition of the shape recovery ratio.

Action 1: the revised text in Line 198 in the revision and Figure 2e (Figure R12a) are as follows:

“The temperature is then gradually increased for free recovery of the programmed TS via Joule heating. During the free recovery process, the angle between two stiff panels gradually increases over time, which is denoted as θ_t and serves as a metric for characterizing the extent of shape recovery. The time-dependent angle θ_t is recorded during the free recovery process and the shape recovery ratio is defined as

$$R_r = (\theta_t - \Delta\theta) / (\theta_{\max} - \Delta\theta), \quad (1)$$

where the numerator represents the recovery angle of the SMP hinge over time while the denominator represents the maximum recovery angle.”

Figure R12 (Figure 2e-g in Manuscript). Shape recovery performance of the shape memory composite. **a** Free recovery process and the definition of the shape recovery ratio of the electrothermal hinged strip structure. **b** Influence of applied current on the shape recovery ratio of the CCF-SMP hinge, and that of the pure-SMP hinge under 0.26 A Joule heating. **c** Influence of applied current on the shape recovery force of the CCF-SMP hinge under constrained recovery, and that of the pure-SMP hinge under 0.26 A Joule heating. Inset: Schematic showing the test method of shape recovery force.

Response 2: the shape recovery force of the SMP hinge is a notable factor that affects the deployment of precisely controlled electrothermal origami (PCEO). As shown in Figure R12c, we compare the maximum shape recovery force of both pure-SMP and CCF-SMP hinges under 0.26 A Joule heating. The maximum shape recovery force of the pure-SMP hinge is only 46.1 mN, significantly smaller than that of the CCF-SMP hinge, which records 188.9 mN. Figure R13 shows snapshots of the electro-induced shape memory process of pure-SMP and CCF-SMP hinged strip structure, where the pure-SMP hinged strip structure cannot overcome its gravity to recover its original shape due to its small shape recovery force. In the revision, we have added text in the Manuscript and Figure 12 in Supplementary Information to explain the impact of shape recovery force on the deployment of PCEO.

Action 2: the revised text in Line 232 in the revision and figure (Supplementary Figure 10) are as follows:

“The recovery force of the pure-SMP hinge is too small to overcome gravity and recover to its permanent shape. The shape recovery process of both the pure-SMP and CCF-SMP hinge can be seen in Supplementary Movie 2.”

Figure R13 (Figure 10 in Supplementary Information). Influence of shape recovery force on the deployment of the PCEO. Snapshots of electro-induced shape memory behavior of pure-SMP and CCF-SMP hinged strip structure. Scale bar, 10 mm.

Response 3: as shown in Figure R12b, a higher current resulted in a faster shape recovery process and a larger shape recovery ratio because a larger current causes CCF-SMP to heat up to the glass transition temperature (T_g) faster. However, the shape recovery force is not directly proportional to the current. Figure R12c presents the maximum recovery force of the CCF-SMP hinge at 0.26 A when most of the CCF-SMP hinge is in the rubbery state. A large proportion of the hinge in the glassy state (0.18 A) or melting state (0.30 A) both lead to a decrease in the shape recovery force. The shape recovery force of the SMP hinge is a notable factor that affects the deployment of PCEO. Figures R12b and R12c depict that the CCF-SMP hinge at 0.30 A presents a faster shape recovery process than at 0.26 A, but a lower recovery force. Compared with 0.26 A, CCF-SMP hinges with applied 0.22 A and 0.18 A not only have a slower recovery speed but also a smaller recovery force. That is why the stimulation of 0.26 A is employed in this work, as it provides the maximum shape recovery force and relatively fast shape recovery. We mentioned this in the Manuscript: “Here, a stimulation current of 0.26 A is employed, as it provides the maximum shape recovery force and fast shape recovery.”

Comment 2.4: Could the shape-reconfigurable PCEO strip structure in Fig 4 transform among configuration 1- 6? And the gripper functional configuration by sequence control seems can only be constructed once?

Response: we thank the reviewer for this valuable question. The shape memory material used in this work is a one-way memory material, which means that the transform among Configurations 1-6 of the PCEO strip structure **needs additional reprogramming**. Similarly, the **reuse of the gripper** functional configuration by sequence control **also requires additional reprogramming**. In the revision, we have added text in the Reconfigurability of the PCEO and Discussion sections of the Manuscript to address the limitations of this system.

Action: the added text in Line 361 and Line 510 in the revision reads:

“It should be noted that all configurations originate from one single PCEO strip, and its reconfigurable deployment requires reprogramming.”

*“However, considerable efforts are needed to translate these design concepts, e.g., the reconfigurable robot gripper and the variable thickness wing, into real-world applications. For instance, **reversible deployment control can be realized by the hybrid-actuation solution including cable driven, etc., as an alternative to the bulky actuation units required by the non-responsive material system.** Moreover, precise control under complicated environments, e.g., dynamic loads and extreme temperatures, deserves further study to align with specific functional requirements.”*

Comment 2.5: The variable thickness wing is a tessellation of five Miura-origami units,

the author should demonstrate the unique advantages and advancements of the proposed wing comparing with the Miura-origami unit in Fig 5.

Response: we thank the reviewer for the constructive comment. The precise control capability of the electrothermal origami allows us to obtain on-demand modulation of **mechanical** and **geometrical** properties. Figure R14a (Figures 5d and 5e in Manuscript) shows the **tunable mechanical** property of a precisely controlled electrothermal Miura-origami (PCEM) by manipulating activation parameters. Figures R14b and R14c (Figures 7c-e in Manuscript) show the **geometrical modulation capability** of PCEM using a variable thickness wing as an example, which is achieved by combining five Miura-origami units with **different geometry designs**. As shown in Figure R14b, we can not only **adjust the deployment angle** of the wing by controlling the heating time but also **adjust the airfoils** by changing the combination of geometric parameters of the five Miura-origami units. We develop a model for the **inverse design of the airfoils** (Figure R14c). Additionally, the variable thickness wing with fiber Miura-origami units also demonstrated that the proposed electrothermal control method can be **extended to complex-shaped origami**, as shown in Figure R15. In the revision, we have added the text and Supplementary Figure 23 (Figure R15) about the complexity of the variable thickness wing.

Action: the added text in Line 462 in the revision and figure (Supplementary Figure 23) are as follows:

*“As illustrated in Fig. 7c, the mode of the variable thickness wing, with a fixed geometric design, can be altered by varying the heating time, **which also demonstrates that the proposed electrothermal control method can be extended to complex-shaped origami.**”*

Figure R14. Comparison of the Miura-origami unit and the variable thickness wing. a (Figures 5d and 5e in Manuscript) PCEM with tunable mechanical properties. **b** (Figures 7c and 7d in Manuscript) Experimental demonstration of different modes of the variable thickness wing obtained by changing heating time, and variation of airfoils of the variable thickness wing with different geometric designs. Scale bar, 40 mm. **c** (Figures 7e and 7f in Manuscript) Design space of the variable thickness wing. Scale bar, 20 mm.

Figure R15 (Figure 23 in Supplementary Information). Construction and dimensions of the variable thickness wing based on PCEM. a Oblique view of the fiber path of the variable thickness wing. The first layer and the last layer are respectively the valley and mountain hinge layer, and the second and fifth layers are CCFs layers, heating for the valley and mountain hinge layer, respectively. **b** Vertical view of fiber path and dimension of the variable thickness wing.

Comments from Reviewer #3

Comment 3.1: The paper presents the creation of origami structures through the use of multi-material additive manufacturing of a shape memory polymer, in this case polylactic acid, and continuous carbon fiber. The carbon fiber material not only acts as a reinforcer, but also as a heating element that can be used to locally heat the shape memory polymer initiating the shape memory process. Advantages of this manufacturing methodology include the ability to stop the shape recovery process by stopping the applied current (thereby enabling the ability to program a multitude of temporary shapes), the ability to move different areas of a larger structure by differing the amount and duration of applied current discretely, and the ability to use the localized deformation capability to locally change the geometry effectively locally controlling the mechanical properties in a metamaterial structure.

Overall, the article is interesting. There is a large amount of supplementary data. I have gone through the supplied documents and videos which are also quite interesting. There are some issues with this article that should be remedied before it can be published. There are some grammatical issues and aspects that could use elaboration, particularly in the Materials and Methods section. I have detailed these issues in line by line manner below:

Response: we thank the reviewer for the positive feedback and the following valuable comments, which help us to further improve the quality of this manuscript.

Comment 3.2: Line 17. First sentence of abstract should be rewritten.

Response: we appreciate the reviewer for the valuable comment. In the revision, we have rewritten the first sentence of the abstract.

Action: the revised text in Line 17 in the revision reads:

“Active origami capable of precise deployment control, enabling on-demand modulation of its properties, is highly desirable in multi-scenario and multi-task applications.”

Comment 3.3: Introduction. There are several other researchers who have made additively manufactured shape memory components with continuous carbon fibers as you have done. Please include references to other works and state how your work differs from theirs.

Response: we thank the reviewers for the constructive suggestion. We have added one paragraph in Introduction to discuss related works about 3D/4D printing of continuous fiber-reinforced shape memory composites to help readers identify how this work is different from prior research.

Action: the added paragraph in Introduction reads:

“Recent advancements in 3D printed continuous fiber-reinforced composites^{31, 32} offer an attractive solution to the issues when using SMP-based 4D printing to realize active origami. These composites demonstrate superior mechanical properties than composites with other reinforcement forms³³⁻³⁶. Additionally, an improved global shape-shifting ability through Joule heating³⁷⁻⁴⁰ of continuous conductive fibers in SMP matrices has been demonstrated with simple structures^{37, 38} or classical lightweight designs, e.g., 2D cellular structures^{39, 40}. However, despite this promising solution, several challenges in the 4D printing of continuous fibers-reinforced composites must be addressed to successfully create active origami with high load-bearing capacity and precise control. Firstly, the design of rigid-foldable origami necessitates a more intricate fiber layout to engineer local stiffness. Moreover, the multi-physical and highly nonlinear deploying process of these composites remains poorly understood. There is also a need to establish related modeling and control methods, which are pivotal for the precise deployment of active origami, ultimately enabling on-demand modulation of its properties for multi-scenario and multi-task applications.”

References:

31. Matsuzaki R, et al. Three-dimensional printing of continuous-fiber composites by in-nozzle impregnation. *Sci Rep* **6**, 23058 (2016).
32. Wang Q, Tian X, Huang L, Li D, Malakhov AV, Polilov AN. Programmable morphing composites with embedded continuous fibers by 4D printing. *Mater Des* **155**, 404-413 (2018).
33. Zeng C, Liu L, Bian W, Leng J, Liu Y. Temperature-dependent mechanical response of 4D printed composite lattice structures reinforced by continuous fiber. *Compos Struct* **280**, 114952 (2022).
34. Dong K, Wang Y, Wang Z, Qiu W, Zheng P, Xiong Y. Reusability and energy absorption behavior of 4D printed continuous fiber-reinforced auxetic composite structures. *Composites, Part A* **169**, 107529 (2023).
35. Zeng C, Liu L, Bian W, Leng J, Liu Y. Compression behavior and energy absorption of 3D printed continuous fiber reinforced composite honeycomb structures with shape memory effects. *Addit Manuf* **38**, 101842 (2021).
36. Dong K, Ke H, Panahi-Sarmad M, Yang T, Huang X, Xiao X. Mechanical properties and shape memory effect of 4D printed cellular structure composite with a novel continuous fiber-reinforced printing path. *Mater Des* **198**, 109303 (2021).
37. Chen H, et al. Electrothermal shape memory behavior and recovery force of four-dimensional printed continuous carbon fiber/polylactic acid composite. *Smart Mater Struct* **30**, 025040 (2021).
38. Zeng C, Liu L, Bian W, Liu Y, Leng J. 4D printed electro-induced continuous carbon fiber reinforced shape memory polymer composites with excellent bending resistance. *Composites, Part B* **194**, 108034 (2020).
39. Ye W, Dou H, Cheng Y, Zhang D. Self-sensing properties of 3D printed continuous carbon fiber-reinforced PLA/TPU honeycomb structures during cyclic compression. *Mater Lett* **317**, 132077 (2022).
40. Dong K, Panahi-Sarmad M, Cui Z, Huang X, Xiao X. Electro-induced shape memory effect of 4D printed auxetic composite using PLA/TPU/CNT filament embedded synergistically with continuous carbon fiber: A theoretical & experimental analysis. *Composites, Part B* **220**, 108994 (2021).

Comment 3.4: Line 103-line 105. It would be useful to state Joule's Law $Q=I^2Rt$ to give an idea of how much heat is being generated over the 30 second interval. It may be better to move Equation (1) on line 164 to earlier in the manuscript.

Response 1: we thank the reviewer for the constructive suggestion. In the revision, we have added the corresponding text.

Action: the added text in Line 112 in the revision reads:

*“Through the infrared image, we observe that the CCF circuit can quickly heat up to 50 °C within 30 seconds through Joule heating (Supplementary Movie 1), **which generates 74.7 J of heat according to Joule’s law.**”*

Response 2: In Figure R16 (Figure 1 in Manuscript), we want to express **what effects will be brought by introducing CCFs into SMP**, such as **conductivity and Joule heating (Figure R16c)**, process control (Figure R16d), and mechanical enhancement (Figure R16e and R18f). However, in Figure R17 (Figure 2 in Manuscript), we want to express **how the introduction of CCFs into SMP will affect** origami structures. Equation (1) provides a good explanation for the heat conduction and dissipation process of CCFs. Therefore, we believe it is most appropriate to place Equation (1) in the Characterization of the shape memory composite Section of the Manuscript.

Figure R16 (Figure 1 in Manuscript). PCEO printed by multifunctional continuous carbon fiber-reinforced SMP 3D printing technology. a Illustration of 3D printing of CCF-SMP PCEO composite. **b** Schematics of the electrothermal hinge enabled by integrating Joule heating CCFs. **c** Demonstration of the electrical conductivity and heating capacity of CCFs. Scale bar, 20 mm. **d** Precise control of the CCF-SMP PCEO showcased by a simple hinged strip structure. **e** Stress-strain curves of the pure-SMP and CCF-SMP at 25 °C and 70 °C. **f** Demonstration of the stiffness improvement of CCF-SMP due to the incorporation of CCFs through experiment and FEA simulation. Scale bars, 20 mm.

Figure R17 (Figure 2 in Manuscript). Heating and mechanical performance of the shape memory composite. **a** Schematic illustration of the electrothermal hinged strip structure and three functions of CCFs (CCFs in the stiff panels and hinges for reinforcement, CCFs in the hinges for heat conduction, and CCFs on the hinges for localized Joule heating). **b** The DMA characterization of pure-SMP and CCF-SMP materials. **c** Influence of applied current on the heating rate of the CCF-SMP hinge. **d** Thermal field distribution of the pure-SMP (without CCFs) and CCF-SMP (with CCFs) hinges under 0.26 A Joule heating for 40 seconds. **e** Shape recovery cycle diagram. **f** Influence of applied current on the shape recovery ratio of the CCF-SMP hinge, and that of the pure-SMP hinge under 0.26 A Joule heating. **g** Influence of applied current on the shape recovery force of the CCF-SMP hinge under constrained recovery, and that of the pure-SMP hinge under 0.26 A Joule heating. Inset: Schematic showing the test method of shape recovery force.

Comment 3.5: Line 149 it please correct $\text{Tan}\delta$ to $\tan \delta$. It may also be useful to indicate that $\tan \delta$ means loss tangent.

Response: we thank the reviewer for the suggestion. In the revision, we have corrected them correspondingly and added an explanation for $\tan \delta$.

Action: the revised text in Line 156 in the revision reads:

“The glass transition temperature (T_g) of the CCF-SMP is identified to be 64.5°C according to the peak of $\tan \delta$ (the ratio between the loss modulus and storage modulus), which is slightly higher than that (63.4°C) of pure-SMP.”

Comment 3.6: Line 183 is somewhat unclear. How is the permanent shape programmed? Generally with SMPs the programmed shape is the permanent shape.

Response: we appreciate the reviewer for this question. Shape memory polymers (SMPs) are polymer networks composed of flexible chain segments subtended across a network of crosslinked netpoints, which **define their permanent shape** (Xia et al., 2022). We can simply understand that the permanent shape is the **stable shape when the SMP is heated** to a temperature above its glass transition temperature (T_g). In this work, the printed shape is the permanent shape. To help the reviewer and reader better understand the programming process of SMP, we have revised the text in the revision.

Action: the revised text in Line 191 in the revision reads:

*“First, one stiff panel is fixed while the other is free and the angle between two stiff panels is denoted as θ . The SMP hinge is heated above its T_g by Joule heating for 30 seconds and then manually deformed from the **permanent shape (PS, as printed shape)** $\theta_{\max} = 180^\circ$ to the temporary shape (TS) $\theta = 0^\circ$, and then allowed to cool by powering off. Once it is cooled to room temperature, the TS is held by loading a 500g weight for 5 minutes, then unloaded, and the programmed TS is fixed with a small bounce-back angle, $\Delta\theta$.”*

References:

Xia X, Spadaccini CM, Greer JR. Responsive materials architected in space and time. *Nat Rev Mater* 7, 683-701 (2022).

Comment 3.7: The sentence says the permanent shape is programmed to the temporary shape. It should be rewritten similar to: the specimen is heated by joule heating and then deformed from the permanent shape $\theta_{\max}=180$ to a temporary shape $\theta=0$ and then allowed to cool by powering off the system.

Response: we thank the reviewer for this helpful comment. In the revision, we have revised the text.

Action: the revised text in Line 190 in the revision reads:

*“First, one stiff panel is fixed while the other is free and the angle between two stiff panels is denoted as θ . **The SMP hinge is heated above its T_g by Joule heating for 30 seconds and then manually deformed from the permanent shape (PS, as printed shape)** $\theta_{\max} = 180^\circ$ to the temporary shape (TS) $\theta = 0^\circ$, and then allowed to cool by powering off. Once it is cooled to room temperature, the TS is held by loading a 500g weight for 5 minutes, then unloaded, and the programmed TS is fixed with a small bounce-back angle, $\Delta\theta$.”*

Comment 3.8: Is the specimen manually deformed to a temporary shape?

Response: we thank the reviewer for this question. All PCEO is **manually** deformed to a temporary shape. In the revision, we have added corresponding explanations.

Action: the revised text in Line 191 in the revision reads:

*“The SMP hinge is heated above its T_g by Joule heating for 30 seconds and then **manually** deformed from the permanent shape (PS, as printed shape) $\theta_{\max} = 180^\circ$ to the temporary shape (TS) $\theta = 0^\circ$, and then allowed to cool by powering off.”*

Comment 3.9: Line 186. Believe the equation for shape fixation should be on its own line and denoted as Equation 2 also the shape recovery equation should be Equation 3. Be sure to correct the numberings of your subsequent equations in the manuscript.

Response: we thank the reviewer for the suggestion. In the revision, we have put the equations of shape fixity (Equation (2)) and shape recovery ratio (Equation (3)) on their own lines. In addition, we have correspondingly corrected the numbering of subsequent equations in the revision.

Comment 3.10: Line 191 I do not think that the equation used to calculate recovery ratio is correct. The first issue I see is that it looks like you are multiplying θ by time. This would effectively give a unit of time in the final value. I believe you are trying to say that the time dependent angle is the angle measured at a discrete time during the recovery process. Am I correct in this assumption? If this is the case, please denote the angle as θt with the t as a subscript. Also please explain what the time dependent angle is more clearly in the text. Figure 2 e indicates it as the measured angle after recovery. If it is the measured angle after recovery, please state this in the text or denote the angle at recovery at the final time or t_f or some other designation that is clear to the reader. Also, the other problem with the equation is that if the angle θ is equal to 1 your recovery ratio will be zero and if the value is greater than 1 your recovery ratio will be negative.

Please check this equation or provide more explanation to this calculation in the text.

Response: we appreciate the reviewer for the constructive comments. The reviewer’s understanding is correct that the time-dependent angle is the angle measured at a discrete during the recovery process. To help the reviewer and reader better understand the definition of the shape recovery ratio. We have revised that the time-dependent angle is denoted as θ_t with t as a subscript and added its explanation in the revision. As shown in Figure R18, we redefine θ_t as the angle between two stiff panels where one stiff panel is fixed while the other is free. During the free recovery process, the

angle θ_t gradually increases over time, which serves as a metric for characterizing the extent of shape recovery. In the revision, we have revised the equation of the shape recovery ratio as $R_r = (\theta_t - \Delta\theta) / (\theta_{\max} - \Delta\theta)$, where the numerator represents the recovery angle of the SMP hinge over time while the denominator represents the maximum recovery angle.

Action: the revised text in Line 200 and Figure 2e (Figure R18) in the revision are as follows:

*“The temperature is then gradually increased for free recovery of the programmed TS via Joule heating. During the free recovery process, **the angle between two stiff panels gradually increases over time, which is denoted as θ_t and serves as a metric for characterizing the extent of shape recovery.** The time-dependent angle θ_t is recorded during the free recovery process and the shape recovery ratio is defined as*

$$R_r = (\theta_t - \Delta\theta) / (\theta_{\max} - \Delta\theta), \quad (2)$$

where the numerator represents the recovery angle of the SMP hinge over time while the denominator represents the maximum recovery angle.”

Figure R18 (Figure 2e in Manuscript). Free recovery process and the definition of the shape recovery ratio of the electrothermal hinged strip structure.

Comment 3.11: The examples provided are complete. The documentation of this work is well done. The reconfiguration capability through localized reprogramming is interesting.

Response: we thank the reviewer again for taking precious time to review our paper and giving us constructive comments and suggestions.

Comment 3.12: Materials and methods section should be written in the past tense. Line 482 should be changed to: The SMP material used was polylactic acid. Similar grammatical issues are found throughout this section. Please correct.

Response: we thank the reviewer for pointing out the grammatical issues. In the revision, we have corrected them correspondingly.

Comment 3.13: Line 483 It is unclear why and how a support material was used during the printing of your specimens. The printer is dual extruder where one extruder printed PLA and the other the continuous carbon fiber. How was the material removed?

Response: we appreciate the reviewer for raising this important question. As shown in Figure R19, we have provided an explanation for the utilization of support material in the printing process of PCEO. Figure R19a shows that this strip origami structure has two valley hinges and one mountain hinge, with **the mountain hinge suspended** in the printing direction. Therefore, **support material is needed to assist in printing the mountain hinge**. Here, we used polyamide (PA) as support material (black material in Figure R19b and R21c), which has a poor interface with SMP. So, it can be peeled off easily after printing. In this work, we **removed the support material with sharp-nose pliers** after printing. Figure R19c depicted that the support material effectively assists in the printing of the strip origami structure. We have added Supplementary Figure 24 (Figure R19) and more detailed explanations for the support material in the revision.

The 3D printer used in this study was a dual-nozzle composite printer, where the left nozzle supplies thermoplastics, such as PLA, and PA, while the right nozzle feeds continuous carbon fibers impregnated with PLA. We can easily replace the materials for the left nozzle through the printer's load/unload program.

Action: the revised text in Line 520 in the revision and figure (Supplementary Figure 24) are as follows:

“The support material polyamide (PA) filament (PolyMide CoPA, Polymaker, China) was used for the printing of PCEO's hanging mountain hinge, which is easy to peel off after printing (Supplementary Fig. 24).”

Figure R19 (Figure 24 in Supplementary Information). Description of supporting materials for origami composite printing. a Vertical and front view of the crease pattern and the computer-aided design of a strip origami structure. **b** Snapshot of the strip origami structure as printed. Scale bar, 20 mm. **c** Snapshots of the strip origami

structure for different views. Scale bar, 20 mm.

Comment 3.14: Line 490 “built” should be changed to “build”

Line 491. Should be changed to 297 mm x 201 mm x 147 mm.

Line 500 should be changed to : was used to record. There are several tense errors in this section as I have mentioned before.

Line 511 Again I think it is correct to write out as 250 mm x 15 mm by 1 mm. Please confirm journal style.

Response: we thank the reviewer again for pointing out the grammatical issues. In the revision, we have corrected them correspondingly.

Comment 3.15: Line 524 sometimes you refer to a figure in text as Figure x, here you call it Fig. 1f. Please check document for consistency.

Response: we thank the reviewer for this comment. We carefully reviewed several articles published in *Nature Communications* and found that when the referenced figure is at the beginning of a sentence, it is expressed as Figure X, while in other cases it is expressed as Fig. X.

Comment 3.16: Lines 552 to Lines 555. Similar to previous comments, it is unclear if the specimen is manually moved to the temporary shape. How is the temporary shape held for five minutes? Is there a measured load applied to the specimen to hold it in this temporary shape. Please elaborate.

Response: we thank the reviewer for this valuable comment. As shown in the response to **Comment 3.8**, all PCEO is initially heated by Joule heating and then manually deformed to a temporary shape. Next, the structure is cooled by powering off. To maintain the temporary shape, we use a 500 g weight to load for 5 minutes, followed by unloading, thus fixing the deformed temporary shape. In the revision, we have added corresponding explanations.

Action: the revised text in Line 191 in the revision reads:

*“First, one stiff panel is fixed while the other is free and the angle between two stiff panels is denoted as θ . The SMP hinge is heated above its T_g by Joule heating for 30 seconds and then **manually** deformed from the permanent shape (PS, as printed shape) $\theta_{\max} = 180^\circ$ to the temporary shape (TS) $\theta = 0^\circ$, and then allowed to cool by powering off. Once it is cooled to room temperature, the TS is held by **loading a 500g weight** for 5 minutes, then unloaded, and the programmed TS is fixed with a small bounce-back angle, $\Delta\theta$.”*

REVIEWERS' COMMENTS

Reviewer #1 (Remarks to the Author):

The revision has improved the clarity of the paper and answered questions from the reviewer in the rebuttal document. While the response in the rebuttal document is satisfactory, some are not reflected in the revised manuscript and SI. Please include them in the updated SI and the manuscript before the paper is published. The reviewer suggest the paper to be published after including these items from the rebuttal document to the updated SI and manuscript.

- Please consider including Fig R5 and R6 from the rebuttal document into the SI. In a way, these two figures are the most significant items for this paper because they highlight the procedure of deriving and using the proposed method. Including simplified versions of them in the main text can also be helpful.

- While the demonstrated example in rebuttal document Fig. R7 provides a PCEO with controller integrated, this example is not included in the updated manuscript nor reflected in the SI. Please consider providing a supplementary video and/or a supplementary figure in SI.

Reviewer #2 (Remarks to the Author):

There are still some issues that need further explanation before publication.

(1) Since reconfigurable PCEO needs additional preprogramming, what is the technical difference between PCEO strip structure and a single PCEO strip?

(2) The variable thickness wing seems also perform one-way deformation, which is rare in real engineering applications. The advantages should be elaborated.

(3) Has the authors considered the deformation perpendicular to XY plane of the variable thickness wing?

Reviewer #3 (Remarks to the Author):

The authors have adequately responded to my comments and concerns. Specifically, errors related to the equations used to calculate the shape fixation ratio and shape recovery ratio have been fixed and the equations now make sense.

Additionally, ambiguities related to the experimental procedure have been clarified.

One minor change I would recommend:

Line 195: please change shape fixity to shape fixation ratio. In some circles fixity is considered an improper way to refer to this shape memory property. Shape fixation ratio is more proper.

Comments from Reviewer #1

Comment 1.1: The revision has improved the clarity of the paper and answered questions from the reviewer in the rebuttal document. While the response in the rebuttal document is satisfactory, some are not reflected in the revised manuscript and SI. Please include them in the updated SI and the manuscript before the paper is published. The reviewer suggest the paper to be published after including these items from the rebuttal document to the updated SI and manuscript.

Response: we thank the reviewer for taking precious time to review our paper and offering constructive comments and suggestions. In the revision, we have added corresponding items to the Supplementary Information and Manuscript.

Comment 1.2: Please consider including Fig R5 and R6 from the rebuttal document into the SI. In a way, these two figures are the most significant items for this paper because they highlight the procedure of deriving and using the proposed method. Including simplified versions of them in the main text can also be helpful.

Response 1: we appreciate the reviewer for this helpful comment. We have added corresponding Figures to Supplementary Information and added corresponding text to Manuscript.

Action: the added text in Line 308 in the revision and figures (Supplementary Figures 17 and 18) are as follows:

“More details about the establishment and guidance of the multi-physic model for PCEO are presented in Supplementary Fig. 17, and the procedure of simulation can be seen in Supplementary Fig. 18.”

Figure R1 (Figure 17 in Supplementary Information). Establishment and guidance of the multi-physics model for precisely controlled electrothermal origami. a Numerical calculation of the highly nonlinear heating process of CCFs. **b** Establishment of the multi-branch thermoviscoelastic model for shape memory composites and the determination of the material parameters. **c** Import of the body heat flux of numerical calculation of CCFs and the material model, establishment of the multi-physics model, and simulation of the shape-shifting behaviors of electrothermal origami with different inputs. **d** Simulation results of the locked shape of the electrothermal origami with a heating time from 0 s to 40 s with a time interval of 2.5 s, and the establishment of the guiding model for the heating time corresponding to the locking angle. **e** Precise control of the airplane-shaped origami with complex deploying routines and the strip origami with reconfigurability guided by the guiding model.

Figure R2 (Figure 18 in Supplementary Information). Flowchart of multi-physics simulation for precisely controlled electrothermal origami. The process includes nine steps: i) measuring the necessary parameters of CCFs for numerical calculation of its heating curve; ii) calculating the body heat flux of CCFs based on their heating curves under different currents; iii) importing the body heat flux of CCFs into the finite element model established for the heat transfer simulation of the electrothermal origami, and determining the boundary conditions (convection coefficients) of the model; iv) validating the effectiveness of the heat transfer model by comparing the hinge heating results of simulation and experiment under different currents; v) importing the multi-branch thermoviscoelastic model for shape memory composites into the simulation model, and creating the shape memory composites programming process; vi) validating the multi-physics model by comparing the shape recovery results of simulation and experiment; vii) simulating the shape-shifting behavior of the electrothermal origami under different heating times; viii) generating the guiding model for the heating time corresponding to the locking angle through simulation, and validating its effectiveness by conducting experiments; ix) guiding the precise control of the electrothermal origami by validated model.

Response 2: in addition, we do include simplified versions of the establishment and simulation procedure of the multi-physics model in the main text. As shown in Figure R3 (Figure 3 in Manuscript), the multi-physics model considers electrical, thermal, and mechanical effects (Figure R3a), and the thermoviscoelastic effect of materials (Figure R3b). Through the simulation and experiment, we investigate the influence of heating time on the shape-shifting behavior of PCEO (Figure R3c), build a guiding model and validate its effectiveness (Figure R3d). Moreover, an airplane-shaped origami is designed as an example to further exhibit the precise controllability of the proposed electrothermal origami. Specifically, more details about the measurement and establishment of material and model parameters are adequately presented in Supplementary Information.

Figure R3 (Figure 3 in Manuscript). Precisely control of electrothermal origami realized by FEA and experiment. a The multi-physics model of the proposed PCEO. **b** Multi-branch thermoviscoelastic model. **c** Simulated temperature distribution and the unfolding angle at power off and after convection cooling for 10 seconds of the CCF-SMP hinge under different heating times. **d** Influence of heating time on the unfolding

angle of the CCF-SMP hinge under 0.26 A Joule heating, three tests are conducted on samples for each data point, and the error bars present the standard deviation of the three repeated data. **e** The crease pattern of the airplane-shaped PCEO structure. **f** Programming of the temporary shape and actively controlled recovery of the PS of the airplane-shaped PCEO structure (LS1, LS2, and LS3 achieved by connecting both end electrodes of hinges 1, 3, and 1-2-3 at a time) realized by FEA. **g** Snapshots of actively controlled recovery of the PS of the airplane-shaped PCEO structure. Scale bars, 40 mm.

Comment 1.3: While the demonstrated example in rebuttal document Fig. R7 provides a PCEO with controller integrated, this example is not included in the updated manuscript nor reflected in the SI. Please consider providing a supplementary video and/or a supplementary figure in SI.

Response: we thank the reviewers for this suggestion. We have added Figure R4 (Figure R7 in the first-round review) to Supplementary Information and added corresponding text to Manuscript.

Action: the added text in Line 356 in the revision and figure (Supplementary Figures 21) are as follows:

“An automatic control and deployment process of PCEO is exemplified and illustrated in Supplementary Fig.21 and Movie 4.”

Figure R4 (Figure 21 in Supplementary Information). An automatic control method of the reconfigurable PCEO. a Control system of the reconfigurable PCEO. The control system consists of two parts: the power supply and control systems. The voltage was provided by a regulated a regulated 24-volt supply (GPD-4303S, GWinstek, China). The switches of the hinges are controlled by the signal from the microcontroller (Arduino UNO R3, DFROBOT, China) via relays (TOUGLESY, CHNT, China). **b** Deployment process of PCEO automatically realized by the self-built control system, Scale bar, 20 mm. Though the control method is out of the scope of this work, we demonstrate the feasibility of realizing automatic control of the reconfigurable PCEO by using this self-built control system. In this work, we chose a manual control method based on removing the alligator clip and manually reconnecting it with other wires, as presented in Supplementary Movie 4.

Comments from Reviewer #2

Comment 2.1: There are still some issues that need further explanation before publication.

Response: we thank the reviewer for taking the time to review our manuscript and

providing helpful comments.

Comment 2.2: Since reconfigurable PCEO needs additional preprogramming, what is the technical difference between PCEO strip structure and a single PCEO strip?

Response: we appreciate the reviewer's question. The PCEO strip structure (with seven hinges) and the single PCEO strip structure (with one hinge) are technically the same. The PCEO strip structure with seven hinges is designed as an example to demonstrate the reconfigurability, spatiotemporal controllability, and additional functionality of PCEO.

Comment 2.3: The variable thickness wing seems also perform one-way deformation, which is rare in real engineering applications. The advantages should be elaborated.

Response 1: we appreciate the reviewer for this important question. We include texts to discuss the limitation of the variable thickness wing, as stated in the Discussion section of the Manuscript: “However, **considerable efforts are needed to translate these design concepts**, e.g., the reconfigurable robot gripper and **the variable thickness wing, into real-world applications**. For instance, **reversible deployment control can be realized by the hybrid-actuation solution including cable driven, etc., as an alternative to the bulky actuation units required by the non-responsive material system**.” The variable thickness wing is designed as an example to exhibit the **geometrical modulation capability** of the electrothermal origami, which has **reconfigurable and controllable geometry**, and **inversely designable airfoil**. Also, the variable wing demonstrates that the proposed electrothermal control method can be **extended to complex-shaped origami**. In the revision, we have added texts to elaborate on its advantages.

Action 1: the added texts in Line 458 and Line 468 in the revision reads:

“The spatiotemporal controllability of the PCEM allows for the design of a variable thickness wing with reconfigurable geometry and inversely designable airfoil for multiple flight scenarios by combining the deployable Miura-origami units with different geometry designs.”

“..., which also demonstrates that the proposed electrothermal control method can be extended to complex-shaped origami.”

Comment 2.4: Has the authors considered the deformation perpendicular to XY plane of the variable thickness wing?

Response: we thank the reviewer for raising this interesting question. The deformation perpendicular to the XY plane of the morphing wing is also important for controlling

the lift-to-drag ratio during flight. However, the focus of the example of variable thickness wings in this work is to demonstrate the geometric on-demand modulation capability of PCEO. Advancing the integration of PCEO with morphing wings constitutes a pivotal area for future research.

Comments from Reviewer #3

Comment 3.1: The authors have adequately responded to my comments and concerns. Specifically, errors related to the equations used to calculate the shape fixation ratio and shape recovery ratio have been fixed and the equations now make sense. Additionally, ambiguities related to the experimental procedure have been clarified.

Response: we thank the reviewer for taking the time to carefully review our paper and helping us improve the paper quality.

Comment 3.2: One minor change I would recommend: Line 195: please change shape fixity to shape fixation ratio. In some circles fixity is considered an improper way to refer to this shape memory property. Shape fixation ratio is more proper.

Response: we appreciate the reviewer for the valuable comment. In the revision, we have changed shape fixity to shape fixation ratio.

Action: the revised text in Line 196 in the revision and Supplementary Figure 6 (Figure R5) are as follows:

“The shape fixation ratio is defined as $R_f = (\theta_{\max} - \Delta\theta) / \theta_{\max}$. The results show that the shape fixation ratio increased with higher current levels, ranging from 88.3% at 0.18 A to 97.6% at 0.30 A (Supplementary Fig. 6).”

Figure R5 (Figure 6 in Supplementary Information). Shape fixation ratio of CCF-SMP hinged strip with different applied currents. Three specimens are tested for each kind of test to obtain an average value of the targeted properties.